# Impact of convectively lofted ice on the seasonal cycle of water vapor in the tropical tropopause layer

Xun Wang[1], Andrew E. Dessler[1], Mark R. Schoeberl[2], Wandi Yu[1], and Tao Wang[3]

[1]Department of Atmospheric Sciences, Texas A&M University, College Station, TX, USA
[2]Science and Technology Corporation, Columbia, MD, USA
[3]University of Maryland, College Park, MD, USA

*Correspondence to*: Andrew E. Dessler (adessler@tamu.edu)

**Abstract.** We use a forward Lagrangian trajectory model to diagnose mechanisms that produce water vapor seasonal cycle observed by the Microwave Limb Sounder (MLS) and reproduced by the Goddard Earth Observing System Chemistry Climate Model (GEOSCCM) in the tropical tropopause layer (TTL). We confirm in both the MLS and GEOSCCM that the seasonal cycle of water vapor entering the stratosphere is primarily determined by the seasonal cycle of TTL temperatures. However, we find that the seasonal cycle of temperature predicts a smaller seasonal cycle of TTL water vapor between 10°N-40°N than observed by MLS or simulated by the GEOSCCM. Our analysis of the GEOSCCM shows that including evaporation of convective ice in the trajectory model increases both the simulated maximum value of the 100-hPa 10°N-40°N water vapor seasonal cycle as well as increasing the seasonal-cycle amplitude. We conclude that the moistening effect from convective ice evaporation in the TTL plays a key role regulating and maintaining seasonal cycle of water vapor in the TTL. Most of the convective moistening in the 10°N-40°N range comes from convective ice evaporation occurring at the same latitudes. A small contribution to the moistening comes from convective ice evaporation occurring between 10°S-10°N. Within the 10°N-40°N band, the Asian monsoon region is the most important region for convective moistening by ice evaporation during boreal summer and autumn.

## 1 Introduction

Stratospheric water vapor is important for the radiative budget of the atmosphere and the regulation of stratospheric ozone (e.g., Solomon et al., 1986; Dvortsov and Solomon, 2001). One of the key features of the tropical lower stratospheric (LS) water vapor is its seasonal cycle often referred to as the "tape recorder" (Mote et al., 1995, 1996). The amount of water vapor entering the stratosphere and its seasonal cycle is primarily controlled by temperatures in the tropical tropopause layer (TTL) (Brewer, 1949; Holton et al., 1995; Fueglistaler et al., 2009). The low TTL temperatures freeze-dry the air, reducing the water vapor mixing ratios and imprinting the seasonal cycle on air ascending into the stratosphere through the TTL (e.g., Mote et al., 1996; Fueglistaler, 2005; Schoeberl et al., 2008; Fueglistaler et al., 2009).

Analyses of observations have suggested that deep convection reaching the TTL may also be important for regulating the

amount of water vapor entering the stratosphere. Nielsen et al. (2007) and Corti et al. (2008) suggested that deep penetrating
convection deposits ice particles above the cold point tropopause, where ice may evaporate and cause a moistening effect.
This idea is also supported by observations of enrichment of the deuterated isotopologue of water vapor (HDO) in the
tropical LS (Moyer et al., 1996; Dessler et al., 2007; Steinwagner et al., 2010). The role of convective ice evaporation in the
stratospheric entry water vapor has also been addressed in several model studies. Schoeberl et al. (2014, 2018, 2019)
quantified the global impact of convective ice on winter 2008/2009 water vapor between 18-30 km, and concluded that, for
global average water vapor between 18-30 km during winter, the convective ice evaporation plays a small role, since
convection rarely reach the level of the tropopause cold point. During El Niño events, convective ice evaporation appears to
play a larger role in the interannual variability of TTL and LS water vapor (Avery et al., 2017; Ye et al., 2018). On longer
time scales, convective ice evaporation was found to contribute to an important fraction of the increase of stratospheric entry
water vapor over the next century in two chemistry-climate models (Dessler et al., 2016).
The goal of this paper is to investigate the impact of convective moistening on the seasonal cycle of water vapor entering the
stratosphere.  Previous analyses have separately investigated the winter/summer impact, interannual variability, and the long-
term trend (Ueyama et al., 2015, 2018; Dessler et al., 2016; Avery et al., 2017; Schoeberl et al., 2014, 2018; Ye et al., 2018).
However, less work has been done on understanding the impact of convective ice on the seasonal cycle. The basics of the
water vapor seasonal cycle can be understood simply: more water vapor enters the LS during boreal summer, when TTL
temperatures are generally higher and vice versa during boreal winter. Observations (Fig. 1a-c) reveal that zonal mean water
vapor is observed to have a larger amplitude seasonal cycle in the NH subtropics near the tropopause level (e.g., Rosenlof,
1997; Randel et al., 1998, 2001, and references therein), despite the fact that the temperature seasonal cycle is symmetric
about the equator (Figs. 1b-c). We will refer to this as "the hemispheric asymmetry". At higher altitudes, the hemispheric
asymmetry gradually disappears (Fig. 1a) (e.g., Randel et al., 1998, 2001).
Previous studies have suggested that this hemispheric asymmetry structure in the water vapor seasonal cycle is due to
processes within the Southeast Asian monsoon and North American monsoon region, including both diabatic and adiabatic
transport in the TTL (Rosenlof, 1997; Randel et al., 1998; Dethof et al., 1999; Bannister et al., 2004; Gettelman et al., 2004;
Pan et al., 1997, 2000, Park et al., 2004, 2007; Wright et al., 2011; Ploeger et al., 2013). Indeed, the MLS data (Fig. 2c) show
that the summertime maxima of the 100-hPa water vapor is confined in the Asian monsoon and North American monsoon
anti-cyclones (Rosenlof, 1997; Jackson et al., 1998; Randel et al., 1998, 2001; Dessler and Sherwood, 2004; Randel and
Park, 2006; Park et al., 2007; Bian et al., 2012) and become weaker above 100 hPa (Figs. 2a-b).
Many previous studies have investigated impact of convection within the monsoon regions on the budget of the stratospheric
entry water vapor. Dessler and Sherwood (2004) used a budget model with and without convection and concluded that,
during summer, moistening by deep penetrating convection increases the northern hemisphere (NH) extratropical water
vapor at 380-K isentrope by 40%. Fu et al. (2006) suggested that the deep convection over the Tibetan Plateau acts as a short

circuit of water vapor ascending across the tropical tropopause. James et al. (2008) used a trajectory model and concluded that air parcels are lifted by convection over Southeast Asia and then transported into the TTL by the monsoon anticyclone, avoiding the cold pool in the deep tropics. However, they pointed out that direct convective injection has a limited impact on the 100-hPa water vapor budget, contributing to 0.3 ppmv of the water vapor in the Asian monsoon region. Schwartz et al. (2013) provided evidence of occasional enhanced 100-hPa and 82.5-hPa water vapor by convective injection over the Asian and North American monsoon regions using satellite observations. Randel et al. (2015) investigated subseasonal variations in 100-hPa water vapor in NH monsoon regions and suggested that stronger convection leads to lower TTL temperatures in the monsoon regions, which results in less LS water vapor there, thereby concluding that the LS water vapor in the monsoon regions is mainly controlled by large-scale transport and TTL temperatures there. Ueyama et al. (2018) investigated the convective moistening effect on 100-hPa water vapor during boreal summer. They used a trajectory model that includes cloud formation, gravity waves, and convective moistening and concluded that convection moistens the water vapor averaged over 10°S-50°N by 0.6 ppmv (~15%) and that convective moistening over the Asian monsoon region plays an important role.

The role of convective ice evaporation in the TTL during boreal summer is still under debate. Furthermore, its impact on the TTL water vapor seasonal cycle has not been fully explored. In this study, we quantitatively investigate the impact of convective ice evaporation on the seasonal cycle of water vapor in the TTL.

**2 Models and Data**

**2.1 MLS water vapor**

We analyze here version 4.2 level 2 water vapor retrieved from the Earth Observing System (EOS) Microwave Limb Sounder (MLS) instrument on the Aura spacecraft (Livesey et al., 2017). Since August 2004, the MLS provides ~3500 vertical scans of the earth's limb from the surface to 90 km each day, covering a latitude range of 82°S to 82°N with a horizontal resolution of 1.5°along the orbit track (Lambert et al., 2007). The MLS water vapor retrieval has a vertical resolution of about 3 km in the TTL, with precision at 100 hPa and 82.5 hPa of 15% and 7%, respectively. The accuracy of the water vapor at 100 hPa and 82.5 hPa is 8% and 9%, respectively (Livesey et al., 2017). We composite the daily standard water vapor between August 2004 to October 2018 to produce monthly means on a horizontal grid of 4° latitude by 8° longitude following the data-screening in Livesey et al., (2017).

**2.2 Ice Water Content from Cloud-Aerosol Lidar with Orthogonal Polarization**

The Cloud-Aerosol Lidar with Orthogonal Polarization (CALIOP) is a two-wavelength polarization elastic backscatter lidar that detects global tropospheric and lower stratospheric aerosol and cloud profiles (Hu et al., 2009; Liu et al., 2009; Vaughan et al., 2009; Winker et al., 2009, 2010; Young and Vaughan, 2009; Avery et al., 2012; Heymsfield et al., 2014). We use the

CALIOP Level 2 Cloud Profile Product in version 4.2, with horizontal resolution of 5 km along-track and 60 m vertically in
the TTL and LS. The CALIOP cloud Ice Water Content (IWC) is derived from a parameterized function of the CALIOP 532
nm cloud particle extinction profiles (Avery et al., 2012; Heymsfield et al., 2014). We use the IWC from all clouds minus the
IWC from thin cirrus clouds (clouds that are not opaque) above 146 hPa, which is a rough estimate of convective ice in the
TTL region, since the CALIOP does not separate convective from non-convective IWC measurements. These CALIOP IWC
data, obtained between May 2008 and December 2013, are then monthly averaged onto the same horizontal and vertical
grids as were used for the MLS data.
**2.3 GEOSCCM**
We also analyze simulations from the Goddard Earth Observing System Chemistry Climate Model (GEOSCCM). The
GEOSCCM couples the GEOS-5 general circulation model (Rienecker et al., 2008; Molod et al., 2012) to a comprehensive
stratospheric chemistry module (Oman and Douglass, 2014; Pawson et al., 2008). The GEOSCCM uses a single-moment
cloud microphysics scheme (Bacmeister et al., 2006; Barahona et al., 2014). The run analyzed here starts in 1998 and ends in
2099 and driven by the Representative Concentration Pathway (RCP) 6.0 greenhouse gas scenario (Van Vuuren et al., 2011)
and the A1 scenario for ozone depleting substances (World Meteorological Organization, 2011). Sea surface temperatures
and sea ice concentrations were prescribed from Community Earth System Model version 1 simulations (Gent et al., 2011).
The model has a horizontal resolution of 2°latitude by 2.5°longitude and 72 vertical levels up to the model top at 0.01 hPa
(Molod et al., 2012).
**2.4 Trajectory Model**
We also use the forward, domain filling, diabatic trajectory model described in Schoeberl and Dessler (2011) and updated in
subsequent publications. The trajectory model uses 6-hourly instantaneous horizontal winds and 6-hourly average diabatic
heating rates to advect parcels using the Bowman trajectory code (Bowman, 1993; Bowman and Carrie, 2002).
Meteorological fields used to drive the model in this paper come from the European Centre for Medium-Range Weather
Forecasts (ECMWF) ERA-interim (ERAi), and Modern-Era Retrospective analysis for Research and Applications-2
(MERRA-2) (Molod et al., 2015; Gelaro et al., 2017), and the GEOSCCM.
In this study, the trajectory model initializes 1350 parcels daily in the upper troposphere on an equal area longitude-latitude
grid covering 0-360°longitude and ± 60°latitude, and with initial water vapor mixing ratio of 200 parts per million by volume
(ppmv). This value is well above saturation, so the parcels are dehydrated to saturation after the first time step of the
trajectory model run.  Sensitivity tests show that our results are not impacted by the initialization values.
The initialization level is at 360-K potential temperature, which is above the average level of zero heating (~355-360 K)
(Fueglistaler et al., 2009) but below the tropical cold point. In the MERRA-2, the average heating rates below ~365 K in the
NH subtropics are negative during boreal summer (not shown), which results in parcels in that region immediately
descending back to the troposphere after initialization. To deal with this problem, we initialize parcels at 360 K in MERRA-
2 simulations. But if the local heating rate at 360 K is negative, we raise the initialization level to the lowest isentropic level
with positive heating rate at the same horizontal position. However, we note that the level of zero heating rate is higher
(~370 K) over the NH monsoon regions in MERRA-2. Releasing parcels at ~370 K over the NH monsoon regions results in
insufficient dehydration and a moist bias there (Schoeberl et al., 2013; Ueyama et al., 2018). To avoid this bias, we set the
local initialization level to 366 K (1 K above the tropical average level of zero heating rate) for those parcels. At the end of
each day, parcels below the 250-hPa pressure surface or above the 5000-K isentrope are removed because they are
considered outside of the model boundaries. We note that the parcels initialized at mid-latitudes mostly descend into the
troposphere.
Along each trajectory, an instant dehydration scheme is used. In this scheme, anytime the relative humidity (hereafter RH,
always with respect to ice) exceeds the dehydration threshold, water vapor is instantly removed to reduce the parcel's RH to
the dehydration threshold. The RH calculation uses 6-hourly temperatures linearly interpolated in time and space to parcel
locations at each time step; the RH is computed using the saturation mixing ratio at that temperature Murphy and Koop
(2005). The pre-set dehydration threshold is 100% RH for the ERAi trajectory runs and MERRA-2 trajectory runs. For the
GEOSCCM trajectory runs, the pre-set dehydration threshold is 80% RH, since in the GEOSCCM dehydration occurs when
the grid-average RH is around this value (Molod et al., 2012). The same parameterization for the pre-set RH threshold of
80% was used successfully analyzing the water vapor interannual variability in the GEOSCCM in Dessler et al. ( 2016) and
Ye et al. (2018). We will refer to this version as the "standard" trajectory model – another version that includes ice
evaporation will be introduced later.
As an alternative to instant dehydration we can run a cloud model along the trajectory model, which is described in
Schoeberl et al. (2014). The cloud model triggers ice nucleation at a prescribed nucleation RH (NRH) threshold and the
number of ice particles produced upon nucleation is proportional to the parcel cooling rate using the relationship derived by
Kärcher et al. (2006). The ice mixing ratio is carried with the parcel along with number of crystals and size. Ice crystal
distribution has a single size mode that varies as the parcels grow or sublimate. Gravitational sedimentation reduces the total
ice amount within the parcel. Ice crystals are assumed to be spheres which is reasonable for small crystals in the upper
troposphere (Woods et al., 2018). The cloud model uses a fixed cloud geometrical thickness of 500 m based on the TTL
cloud thickness distribution observed by CALIOP (Schoeberl et al., 2014). We also assume that ice falling out of the cloud
slowly sublimates in sub-saturated layers well below the cloud. The cloud model incorporates more realistic physics than the
instant dehydration scheme we use in the standard trajectory model and it produces good agreement with observational data
from aircraft flights (Schoeberl et al., 2015). The physics in the cloud model has a net effect of slowing down the parcels'
dehydration rate and increasing water vapor in the LS compared to the instant dehydration scheme (Schoeberl et al., 2014).
All the trajectory model runs include methane oxidation as a water source as described in Schoeberl and Dessler (2011), but
this process is unimportant in the TTL and LS. We start all trajectory models on 1/1/2000 and analyze the model results from
2005 to October 2018, so that we can compare the ERAi and MERRA-2 driven trajectory results to the MLS observations.
The GEOSCCM is a free-running model, so interannual variability of the model will not match MLS observations. We will
therefore compare a multi-year average of the GEOSCCM to observations.
**3 Results**
**3.1 Impact of convective moistening on the seasonal cycle**
Figures 1d-i show the water vapor seasonal cycle at 100 hPa, 82.5 hPa, and 68 hPa simulated by the standard trajectory
model driven by ERAi, and MERRA-2 in which dehydration is entirely driven by temperature and there is no convective
influence (See Table 1 for summary of the trajectory model cases). To compare with the MLS, we averaged the trajectory
water vapor fields in the vertical using the MLS averaging kernels following the instructions from Livesey et al. (2017). The
trajectory models fail to produce the hemispheric asymmetry, the larger water vapor seasonal cycle in the NH subtropics in
August-September at 100 hPa and 82.5 hPa (Figs. 1e-f and 1h-i). Specifically, the ERAi and MERRA-2 trajectory models
underestimate the 100-hPa seasonal amplitude over 10°N-40°N by 0.5 ppmv (24%) and 0.89 ppmv (43%) respectively. At
68 hPa, the trajectory models agree with the MLS that the seasonal cycle is approximately centered over the equator,
although they underpredict the MLS (Figs. 1a, 1d, and 1g). During June-July-August (JJA) (Figs. 2d-i), the trajectory models
underestimate the maxima over the Asian and North American monsoon regions (Figs. 2e-f, 2h-i), which agrees with
Ueyama et al. (2018) who also showed that the trajectory model driven by the ERAi without any convective influence fails
to reproduce the boreal summer maxima. At 68 hPa, the monsoonal maxima are nearly gone (Figs. 2d and 2g).
We also ran the ERAi and MERRA-2 simulation with the cloud model described in Section 2.4 operating along the
trajectory model, with 100% NRH (Table 1).  Note that this version of the trajectory model does not have any convective ice
in it, so water vapor is still regulated entirely by TTL temperatures. Figures 1j-o show that the cloud model produces larger
water vapor values in the seasonal cycles at 100 hPa, 82.5 hPa, and 68 hPa. There is also a slight increase in the seasonal
maximum poleward of 20°N (Figs 1l and 1o) at 100 hPa. The ERAi and MERRA-2 trajectory models with the cloud model
increase the 10°N-40°N seasonal amplitude at 100 hPa by 0.1 ppmv (6%) and 0.08 ppmv (7%) - a small improvement
compared to the instant dehydration scheme. However, the cloud model doesn't help reproduce the observed hemispheric
asymmetry in the seasonal cycles at 100 hPa and 82.5 hPa – it basically increases water vapor both north and south of the
equator. During JJA (Figs. 2l and 2o), the cloud model increases the 100-hPa water vapor values over the Asian monsoon
and North American monsoon regions, but there is still an underestimation compared to the MLS. We note that the NRH
threshold of 100% can be too low, since previous observations showed that the NRH can be as high as 160%-170% in the
TTL region during winter (Jensen et al., 2013). Schoeberl et al. (2016) showed that the sensitivity of trajectory simulated
water vapor to the NRH threshold is 0.1-0.2 ppmv per 10 percent NRH, and that an NRH threshold of 140-145% in the
trajectory model produces water vapor in better agreement with the MLS observations during winter. However, our result
regarding the hemispheric asymmetry and boreal summer maxima agrees with Ueyama et al. (2018), who set the NRH
threshold to 160%, indicating that the hemispheric asymmetry is not sensitive to the choice of NRH threshold. Thus,
regardless of dehydration scheme, models that regulate water vapor only through TTL temperatures and large-scale transport
do not reproduce important features of the 100 – 82.5 hPa water vapor seasonal cycle, including the observed hemispheric
asymmetry.
Our hypothesis is that convective moistening is causing the hemispheric asymmetry in the TTL water vapor seasonal cycle.
Previous analyses (e.g., Ueyama et al., 2018) have attempted to test this idea by directly incorporating observed convection
into the trajectory model and then evaluating how agreement with water vapor observations improved.  However, estimating
convective height from passive infrared measurements is difficult and Ueyama et al. (2018) noted that errors in the
convective heights created issues in their analysis. Given this significant uncertainty in an observation-only approach, we
therefore take a different tack. We perform a parallel analysis with the GEOSCCM, a model that we show below reproduces
the hemispheric asymmetry and we will examine the causes of the asymmetry in that model and then evaluate whether we
think that is what is going on the real world.
In our analysis, we first run the standard trajectory model driven by meteorology from the GEOSCCM which, like the
standard models analyzed above, uses instant dehydration to regulate water vapor exclusively through TTL temperatures.
We also run a second version of the trajectory model, the "ice model", in which we add the convective moistening to the
trajectory model.
The GEOSCCM outputs convective ice at every step. To add convective moistening to our trajectory model, we linearly
interpolate the GEOSCCMs' 6-hourly three-dimensional convective ice field to the location and time of each trajectory's
time step. Then, at each time step, we assume complete evaporation of the ice into the sub-saturated parcels by adding the ice
water content to the parcels' water vapor — although we do not let parcels exceed the pre-set RH threshold of 80%. This is
similar to the convective moistening scheme used by Schoeberl et al. (2014), who used MERRA anvil ice to facilitate the
convective moistening in the trajectory model. After each encounter, we do not allow parcels to carry any remaining
convective ice downstream as Schoeberl et al. (2014) did in their ASC case. Ueyama et al. (2018) used a similar convective
moistening scheme, where they saturated the column model up to the observed cloud top when a parcel's trajectory
intersects a convective cloud. Because we assume instant dehydration and instant evaporation of the ice, we consider the
convective moistening in our trajectory model runs to be an upper limit of the impact of convective ice evaporation on the
TTL water content in the GEOSCCM (Dessler et al., 2016).
To test if GEOSCCM convective ice field is realistic, we compare GEOSCCM convective ice with CALIOP ice data (ppmv)
(Figures 3 and 4). For the CALIOP, we show IWC from all clouds minus IWC from thin cirrus clouds (not opaque), above
146 hPa, which is a rough estimate of convective ice in the TTL region, although it is almost certainly an underestimate of
true convective ice amount. There's general agreement between the spatial pattern of GEOSCCM and CALIOP convective
ice. However, the GEOSCCM generally produces more convective ice and higher convective top altitudes than the CALIOP.
To address these problems in the GEOSCCM, we also show the GEOSCCM convective ice field reduced by 80% (0.2ice),
which brings tropical GEOSCCM convective ice into better agreement with the CALIOP values at 121 hPa and above (Figs.
3e-f and 4e-f). We show below two sensitivity tests that show our results are not sensitive to the overestimation of
convective IWC and convective top altitude by the GEOSCCM.
The water vapor seasonal cycles from the GEOSCCM and various GEOSCCM trajectory model runs (Table 1) are shown in
Fig. 5. These have been re-averaged in the vertical using the MLS averaging kernels (Livesey et al., 2017) to facilitate
comparison with MLS. We focus on the 100-hPa level, where the hemispheric asymmetry is strongest. We note that the 100-
hPa level is in the TTL and is not strictly above the tropopause, especially in the summer NH monsoon region. However,
processes on this level play a key role in determining stratospheric water vapor (Fueglistaler et al., 2009).
The GEOSCCM reproduces the hemispheric asymmetry seen in the MLS observations (compare Fig. 5a with Fig. 1c), and
shows that during JJA the 100-hPa water vapor maxima are located over the Asian monsoon and North American monsoon
regions (compare Fig. 5b with Fig. 2c). The standard trajectory model driven by GEOSCCM meteorology, which regulates
water entirely through TTL temperatures, does not reproduce the hemispheric asymmetry (Fig. 5c). That model also
underestimates the JJA water vapor values in the Asian monsoon region and North American monsoon region (Fig. 5d).
These results are similar to the comparison between MLS and the standard trajectory models driven by ERAi and MERRA-2
(Figs. 1f, 1i, 2f, and 2i)
Fig. 6 shows the 100-hPa water vapor seasonal cycles in the NH subtropics (10°N-40°N), deep tropics (10°S-10°N), and
southern hemispheric subtropics (10°S-40°S). To aid in comparison, we have subtracted the annual mean from each data set.
The standard model generally agrees well with the GEOSCCM and MLS in the 10°S-10°N and 10°S-40°S region (Figs. 6b-
d). This suggests that the water vapor seasonal cycle in those regions is mainly controlled by the TTL temperatures and
large-scale transport and implying that other factors, including convective ice evaporation, are less important. In the 10°N-
40°N region, however, the standard model does a poor job, underestimating the MLS and GEOSCCM seasonal amplitude by
1.15 ppmv (55%) and 1.23 ppmv (57%) (Figs. 6a and 6d).
If we add convective ice evaporation to the trajectory model, then the models show a clear hemispheric asymmetry in the
100-hPa water vapor seasonal cycle and more pronounced seasonal maxima over the monsoon regions (Figs. 5e-h). Fig. 6
shows that the ice model and the 0.2ice model (the trajectory model where we add 0.2ice as shown in Figs. 3e-f) produce

boreal summer and autumn water vapor values in the 10°N-40°N much closer to the GEOSCCM and MLS. The ice model and the 0.2ice model increase the 10°N-40°N seasonal maximum by 2.39 ppmv (63%) and 1.65 ppmv (44%), and increase the seasonal amplitude by 1.55 ppmv (169%) and 1.03 ppmv (112%) (Figs. 6a and 6d). This means convective ice evaporation is particularly important to the 100-hPa water vapor mixing ratio in NH subtropics during boreal summer and autumn, thereby playing a key role in the seasonal cycle there.

Fig. 5e shows that our ice model generates too much water vapor, consistent with too much IWC in the TTL. Given that the GEOSCCM's water vapor fields are reasonable (e.g., Fig. 5a vs. Fig. 1c), this further emphasizes that our instant dehydration model evaporates too much water vapor, thus yielding an upper limit of the impact of ice evaporation. It may also indicate a cancelling error in the GEOSCCM: too much water from ice cancelling a too low dehydration threshold (80%). Clearly, more research on this question is warranted.

We ran another GEOSCCM trajectory ice model to test the sensitivity of water vapor at 100 hPa to convective ice altitude. This was in response to our observation that convective ice in the GEOSCCM went too high into the stratosphere compared to the CALIOP (Fig. 3a vs. Fig. 3c), and we wanted to see if this influences our results. In this test run, we do not allow any ice above the 90-hPa surface to evaporate, so that we eliminate any convective influence that is above that altitude. The zonal mean seasonal cycle and JJA water vapor at 100 hPa from this run is shown in Figs. 5i-j. The difference between the seasonal cycles from the ice model and this test run is less than 0.3 ppmv between 30°S – 30°N. The larger moisture difference at higher latitudes comes from convective moistening in the lowermost stratosphere. However, the hemispheric asymmetry is well reproduced by this test run. We thereby conclude that the impact of convective ice evaporation on the 100-hPa water vapor seasonal cycle is insensitive to convective ice occurrence that is too high in altitude.

These results suggest that convective ice evaporation in the TTL is important to the 100-hPa water vapor seasonal cycle in the NH subtropics in the GEOSCCM. Combined with the fact that the GEOSCCM has reasonable water vapor and convective fields, and that our results are insensitive to errors in the IWC amount and convective altitudes in the GEOSCCM, we believe this is a plausible explanation for the hemispheric asymmetry. That plausibility is strongly supported by the lack of a competing hypothesis for the asymmetry.

**3.2 Source regions of convective ice evaporation**

Our result begs the question of which region contributes most to this convective moistening? Here we define the quantity *net convective moistening* to be the water vapor mixing ratio in the ice model minus that in the standard trajectory model. The net convective moistening thus represents the net water vapor added by convective ice evaporation. In this section, we investigate the regional contribution to net convective moistening in the NH subtropics in the GEOSCCM, so we don't use the MLS averaging kernels in the vertical direction, as we did in Section 3.1. The net convective moistening in the NH

subtropics seasonal maximum and seasonal amplitude is therefore 2.69 ppmv and 1.68 ppmv – slightly different from the
values we show in Fig. 5. We also note that the 100-hPa net convective moistening value in the 10°S - 50°N domain
produced by our GEOSCCM analysis during boreal summer is larger than the value of 0.6 ppmv produced by the
observational analysis of Ueyama et al. (2018)'s observational analysis. This is because the combination of instant
dehydration scheme and instant ice evaporation scheme we use in the trajectory model lead to larger net convective
moistening. This also reinforces the idea that the ice model we use in this paper provides an upper limit of the impact of
convective ice evaporation on the 100-hPa water content in the GEOSCCM (Dessler et al., 2016). Thus, we view our results
to mainly be qualitatively useful.
We also quantify the *convective evaporation rate* in the ice model. To do this, we record the location and amount of water
vapor added to each parcel from ice evaporation on every time step. We then grid and average these values to produce a
three-dimensional field of the ice evaporation rate (in units of ppmv day$^{-1}$). Note that water vapor added by convection will
not necessarily make it into the stratosphere — the added water vapor may be removed in subsequent dehydration events.
Fig. 7a and 7b show that convective evaporation rate generally follows the IWC. However, we see that the highest ice
evaporation rates and net convective moistening (Fig. 7d) in regions where IWC is high and RH (Fig. 7c) is low (Dessler and
Sherwood, 2004). In regions where both IWC and RH are high, evaporation is suppressed, and any air that is moistened by
evaporation is rapidly re-dehydrated.
To determine how evaporation in different regions contribute to the 10°N-40°N seasonal cycle, we separately track the
amount of water vapor produced by evaporation in specific latitude bands. Fig. 8 shows the seasonal cycle of net convective
moistening at 100 hPa averaged in the 10°N-40°N region contributed by evaporation of convective ice between 10°N-40°N
and 10°S-10°N. We note that, to obtain the net convective moistening and fractions contributed by specific latitude bands,
we have not subtracted annual mean from the seasonal cycles in this plot like we did in Fig. 6.
During the winter (DJF), contributions from ice evaporation between 10°S-10°N and 10°N-40°N are about even, with
slightly larger contribution from 10°S-10°N. During the summertime (JJA), however, evaporation of convective ice in the
10°N-40°N region is the dominant contributor to the net convective moistening. Specifically, it contributes to 63% (1.7
ppmv) and 59% (0.9 ppmv) of the net convective moistening in the 10°N-40°N water vapor seasonal maximum (September)
value and seasonal amplitude, respectively. Convective ice evaporation between 10°S-10°N plays a smaller role,
contributing to 31% (0.83 ppmv) and 17% (0.28 ppmv).
Next, we investigate net convective moistening in the 100-hPa 10°N-40°N water vapor seasonal cycle contributed by
specific regions within the 10°S-40°N domain. To do this, we divide the 10°S-40°N domain into 12 equal-area boxes. We
average the net convective moistening contributed by each of these boxes using the same method we used to calculate the
contribution by 10°N-40°N and 10°S-10°N. Fig. 9 shows the contribution from each box region to the net convective
moistening in the 100-hPa 10°N-40°N water vapor seasonal maximum value in September and the seasonal amplitude.
We find that contribution from the box regions over Southeast Asia (10°N-40°N, 60°E-120°E), subtropical Western Pacific
(10°N-40°N, 120°E-180°E), and North America (10°N-40°N, 120°W-60°W) dominate. The Southeast Asia region is most
important, contributing to 20% (0.54 ppmv) and 20% (0.3 ppmv) of the net convective moistening in the 10°N-40°N water
vapor seasonal maximum value and seasonal amplitude, respectively. This conclusion is consistent with Ueyama et al.
(2018), who showed that parcels in the 10°S-50°N domain at 100 hPa are mainly hydrated by convection over Southeast
Asia. Specifically, they showed that convection over the Asian monsoon region (0-40°N,40°E-140°E) contributes
approximately 50% of the total convective moistening (10°S-50°N) at 100 hPa during August 2007. We computed the
contribution from the same domain and got a contribution of 36%. The reason we produce a smaller contribution from this
domain is that the GEOSCCM produces more convective ice over the tropical west Indian Ocean (Fig. 4), which results in
larger convective moistening contributed by that region.
The subtropical Western Pacific also contributes to the net convective moistening in the 100-hPa 10°N-40°N water vapor
seasonal cycle. This is due to the abundant convective ice over the subtropical west Pacific (Fig. 4b), which is likely related
to the east-west oscillation of the Asian monsoon anticyclone (Pan et al., 2016; Luo et al., 2018). The North America region
is less important in the ice model, contributing to 12% (0.3 ppmv) and 13% (0.21 ppmv) of the net convective moistening in
the 10°N-40°N water vapor seasonal maximum value and seasonal amplitude. The GEOSCCM underestimates the observed
convective ice over the North American monsoon above 120 hPa (not shown), which may cause the contribution from the
North American region to be underpredicted.

## 4 Summary

In this study, we investigated mechanisms that drive the seasonal cycle of water vapor in the TTL. We use a Lagrangian
trajectory model (Schoeberl and Dessler, 2011) to analyze the seasonal cycle in observations of water vapor made by the
Microwave Limb Sounder (MLS) (Lambert et al., 2007; Livesey et al., 2017) as well as simulated fields from the Goddard
Earth Observing System Chemistry Climate Model (GEOSCCM) (Rienecker et al., 2008; Molod et al., 2012; Pawson et al.,
2008; Oman and Douglass, 2014).
Water vapor's seasonal cycle in the TTL and tropical lower stratosphere (LS), sometimes referred to as the "tape recorder,"
has highest values of water vapor entering the stratosphere during NH summer. We confirm in both the MLS observations
and in the GEOSCCM that this is mainly due to the seasonal cycle of TTL temperatures. However, closer examination of the
data reveals some deficiencies in this simple picture. Both the MLS and GEOSCCM show that the water vapor seasonal
cycle in the TTL has a hemispheric asymmetry, with maximum seasonal cycle between 10°N-40°N, despite the fact that the
TTL temperature seasonal cycle is symmetric about the equator (e.g., Rosenlof, 1997; Randel et al., 1998, 2001, and
references therein). The hemispheric asymmetry is strongest at 100 hPa. Trajectory models that only regulate TTL and
tropical LS water vapor using temperatures (Schoeberl and Dessler, 2011) from ERAi, MERRA-2, and GEOSCCM all
produce weaker water vapor seasonal cycles between 10°N-40°N compared to the MLS and GEOSCCM. These indicate that
the 100-hPa seasonal oscillation between 10°N-40°N is too large to be simply explained by TTL temperatures.
Recent studies suggested that evaporation of convective ice in the TTL also contributes to the amount of water vapor
entering the stratosphere (Nielsen et al., 2007; Corti et al., 2008; Steinwagner et al., 2010; Dessler et al., 2016; Ueyama et
al., 2015, 2018 Schoeberl et al., 2014, 2018; Ye et al., 2018). To better understand this, we analyze a chemistry-climate
model where evaporation of convective ice is known to add water to the TTL (Dessler et al., 2016; Ye et al., 2018). Previous
work (Ye et al., 2018) has shown that the behavior of the GEOSCCM in the TTL is reasonable and agrees well with
observations. Comparisons with Cloud-Aerosol Lidar with Orthogonal Polarization (CALIOP) observations (Hu et al., 2009;
Liu et al., 2009; Vaughan et al., 2009; Winker et al., 2009, 2010; Young and Vaughan, 2009; Avery et al., 2012; Heymsfield
et al., 2014) show that the GEOSCCM IWC has too much ice in the TTL, but we used two sensitivity tests to show that our
results are not sensitive too these disagreements.
Using a version of the trajectory model driven by GEOSCCM meteorology that includes evaporation of GEOSCCM
convective ice, we obtained a more accurately simulated seasonal cycle of the 100-hPa water vapor between 10°N-40°N and
the hemispheric asymmetry compared to the GEOSCCM. We showed results where the GEOSCCM's IWC is reduced to
20% of the original value, and that did not affect our conclusions. In addition, our results are also not sensitive to
GEOSCCM putting convective ice too high in altitude (above 90 hPa). In these runs, adding convective ice to the trajectory
model increases the 100-hPa 10°N-40°N seasonal maximum by 1.65 ppmv (44%), and increases the seasonal amplitude by
1.03 ppmv (112%). We note that our estimate of convective moistening in the NH subtropical seasonal cycle in the
GEOSCCM is larger than the value produced by previous studies based on observations (e.g., Ueyama et al., 2018). This
could be due to overestimates of IWC by the GEOSCCM or because the instant dehydration scheme and instant ice
evaporation scheme we use lead to a greater convective impact on water vapor values overall. Therefore we regard our
results as providing insight for understanding the observations, but we caution against assuming that the numbers we
calculate for ice evaporation in the GEOSCCM are quantitatively accurate estimates of our atmosphere's values.
The majority of the convective moistening at 100-hPa and between 10°N-40°N is contributed by convective ice evaporation
in the 10°N-40°N latitudinal range during boreal summer. The maximum convective ice evaporation in this region is due to
available convective ice and relative humidity low enough to allow it to evaporate (Dessler and Sherwood, 2004). Ice
evaporation between 10°N-40°N contributes to 63% and 59% of the net convective moistening in the 100-hPa 10°N-40°N
water vapor seasonal maximum value and seasonal amplitude. Between 10°N-40°N, the Asian monsoon region plays the
most important role in convective moistening by ice evaporation. Convective ice evaporation in other regions, including the
deep tropics between 10°S-10°N, has a smaller influence in 100-hPa water vapor between 10°N-40°N. However, since the
GEOSCCM underestimates the observed convective ice over the North American monsoon above 120 hPa (not shown), it is
likely that this causes an underestimation of the moistening effect of convective ice over the North American region.
Previous studies showed that the ratio of isotopic water vapor (HDO), an indicator of sublimation of convective ice and in-
mixing (e.g., Dessler et al., 2007; Hanisco et al., 2007; Randel et al., 2012), enhances over the American monsoon region
during boreal summer, suggesting more convective ice evaporation there (Randel et al., 2012). This paper does not discuss
the HDO issue, and more work needs to be done in the future.
To summarize, we find that TTL temperature variations alone cannot explain the seasonal cycle of water vapor at 100 hPa in
MLS observations over the NH subtropics, 10°N-40°N (although temperature does explain the seasonal cycle in the tropics,
10°S-10°N and southern subtropics, 10°S-40°S). To try to understand the other mechanisms at work, we analyze a
chemistry-climate model, the GEOSCCM, which reproduces the MLS observations and has been shown to accurately
simulate the TTL. We find that, in the GEOSCCM, evaporation of convective ice in the TTL is responsible for the larger
seasonal cycle in the 100-hPa NH subtropics. We therefore conclude that evaporation of convective ice in the TTL, mainly
in boreal summer, is the most likely explanation for the observed larger seasonal cycle in the NH subtropics. We concur that
the seasonal cycle of the TTL temperatures is the major driver of the seasonal cycle of water vapor entering the stratosphere,
but we find that the contribution from evaporation of convective ice fills in more details of this simple picture. Our findings
emphasize the need to better understand and quantify the magnitude and spatial pattern of convective ice evaporation in the
TTL.
*Data availability.* The water vapor observed by MLS is available from https://mls.jpl.nasa.gov/. The ice water content
observed by CALIOP is available from https://eosweb.larc.nasa.gov/. The MERRA-2 meteorological fields are available
from        https://disc.gsfc.nasa.gov/.        The       ERAi       meteorological       fields       are       available       from
https://www.ecmwf.int/en/forecasts/datasets/reanalysis-datasets/era-interim.
*Competing interests.* The authors declare that they have no conflict of interest.
*Author contribution.* Xun Wang performed analysis, and wrote the original draft. Andrew E. Dessler provided the
conceptualization, guidance, and editing. Mark R. Schoeberl and Tao Wang contributed to the trajectory model code,
methodology, discussion, and editing. Wandi Yu contributed to methodology and discussion.
*Acknowledgments.* This work was supported by NASA grants NNX16AM15G and 80NSSC18K0134, both to Texas A&M
University. We would like to thank Dr. Luke Oman for providing the GEOSCCM meteorological fields used in this study.

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

**Table 1: Summary of trajectory model cases**

| Trajectory Model Cases | Description |
| --- | --- |
| ERAi standard trajectory model | Instant dehydration with no convective influence |
| MERRA-2 standard trajectory model | Instant dehydration with no convective influence |
| ERAi trajectory with cloud model | Dehydration with the cloud model, but with no convective influence |
| MERRA-2 trajectory with cloud model | Dehydration with the cloud model, but with no convective influence |
| GEOSCCM standard trajectory model | Instant dehydration with no convective influence |
| GEOSCCM ice model | Instant dehydration. Convective ice instantly evaporates to sub-saturated parcels |
| GEOSCCM 0.2ice model | Instant dehydration. GEOSCCM convective ice input is decreased by 80%. Convective ice instantly evaporates to sub-saturated parcels |
| GEOSCCM ice model below 90 hPa | Instant dehydration. Convective ice evaporation above the 90-hPa surface is not allowed |




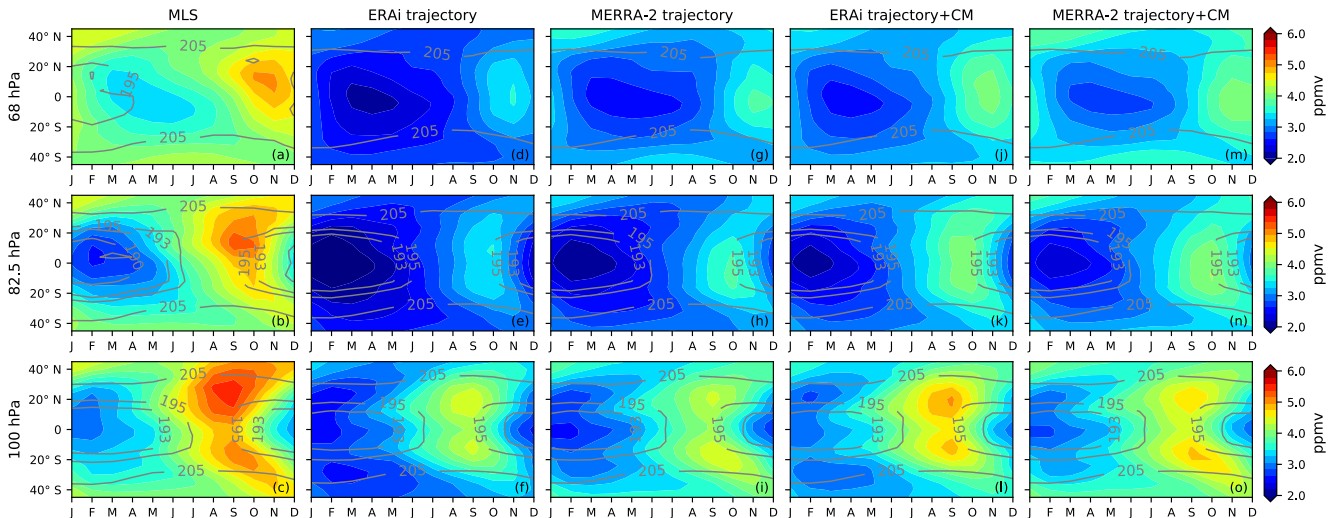

**Figure 1: Zonal mean seasonal cycle water vapor (ppmv, color shading) and temperature (K, contour lines) between 40°S - 40°N**
**from (a)-(c) MLS, (d)-(f) ERAi trajectory model, (g)-(i) MERRA-2 trajectory model, (j)-(l) ERAi trajectory model with the cloud**
**model, and (m)-(o) MERRA-2 trajectory model with the cloud model at 68 hPa (top row), 82.5 hPa (middle row) and 100 hPa**
**(bottom row).**

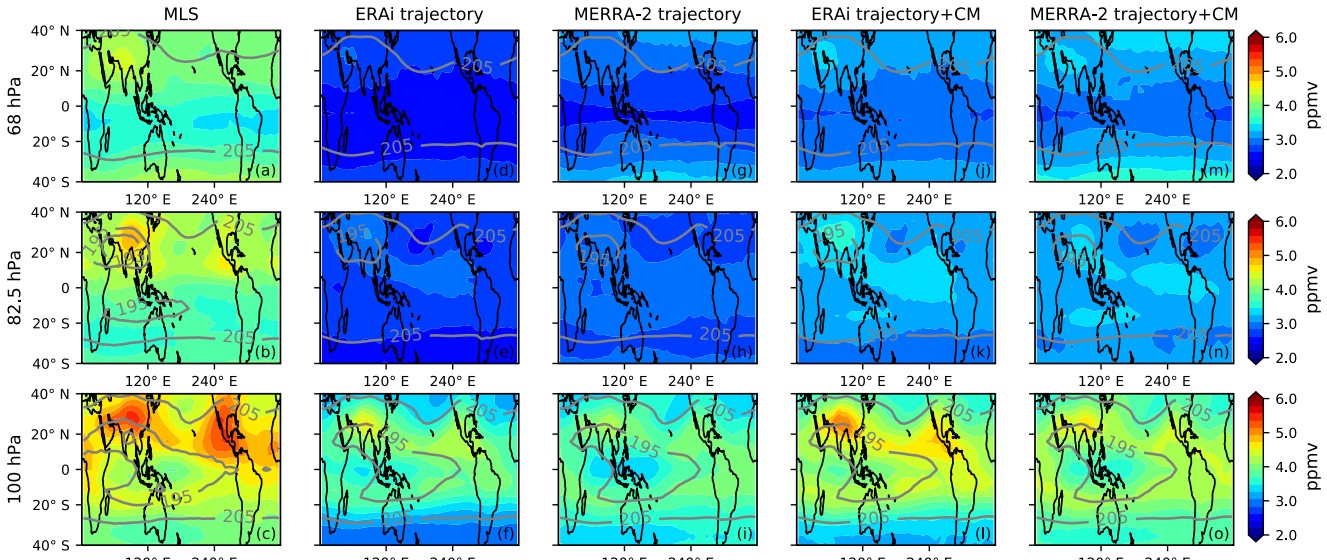

Figure 2: JJA water vapor (ppmv, color shading) and temperature (K, contour lines) between 40°S - 40°N from (a)-(c) MLS, (d)-(f) ERAi trajectory model, (g)-(i) MERRA-2 trajectory model, (j)-(l) ERAi trajectory model with the cloud model, and (m)-(o) MERRA-2 trajectory model with the cloud model at 68 hPa (top row), 82.5 hPa (middle row) and 100 hPa (bottom row).

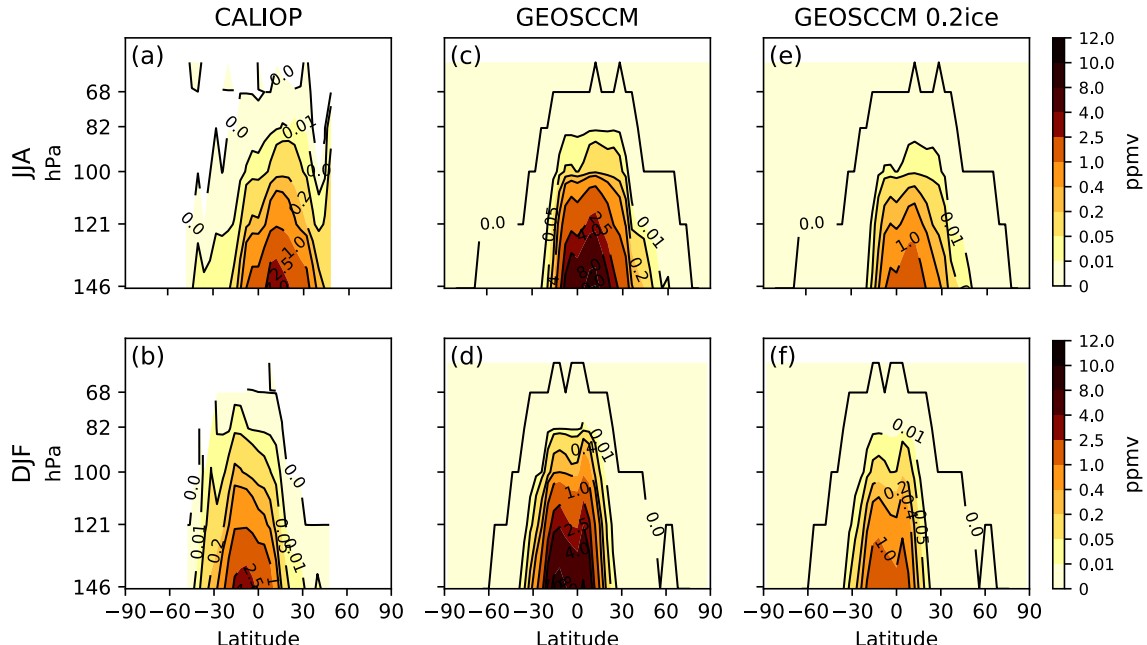

Figure 3. Zonal mean convective IWC (ppmv) from CALIOP and GEOSCCM above 146 hPa. The CALIOP ice data is averaged over JJA (top row) and DJF (bottom row) from 200805 – 201312. For CALIOP data (a and b), we show the ice from all clouds minus the ice from cirrus clouds, which is a rough estimate of convective ice in the TTL region. For GEOSCCM ice, we show convective ice (c and d). To better match the tropical average CALIOP ice field above 120 hPa, we decrease the GEOSCCM ice by 80% (0.2ice) and show them in panels e and f. Note we use a nonlinear color scale.

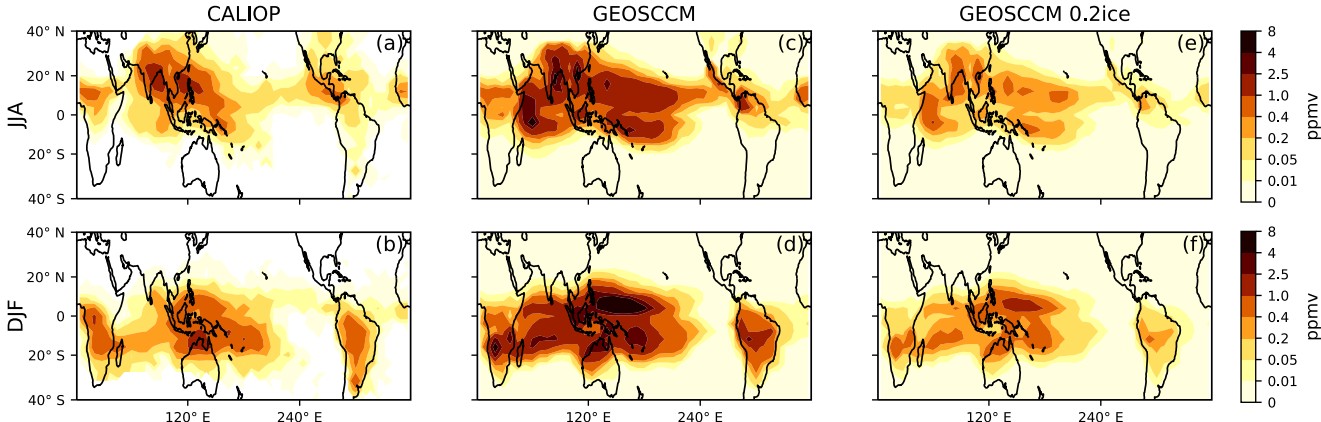

Figure 4. Same as Fig. 3, but for convective IWC (ppmv) averaged between 121-82.5 hPa During JJA and DJF.


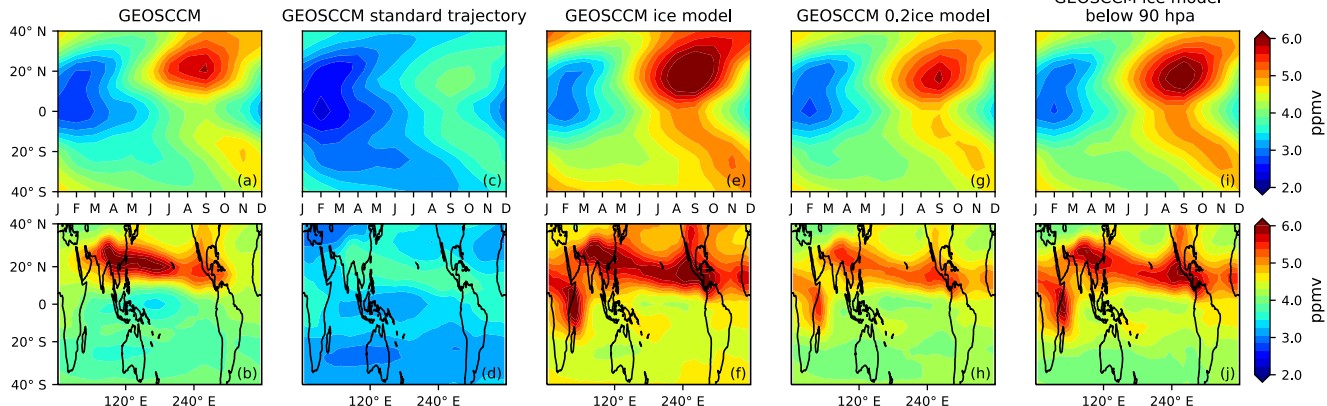

**Figure 5. Top panel: Zonal mean seasonal cycle of 100-hPa water vapor (ppmv) between 40°S - 40°N from (a) GEOSCCM, (c)**
**GEOSCCM standard model, (e) GEOSCCM ice model (g) GEOSCCM 0.2ice model, and (i) GEOSCCM ice model with ice below**
**90 hPa. Bottom panel: JJA 100-hPa water vapor (ppmv) between 40°S - 40°N from (b) GEOSCCM, (d) GEOSCCM standard**
**model, (f) GEOSCCM ice model (h) GEOSCCM 0.2ice model, and (j) GEOSCCM ice model with ice below 90 hPa.**


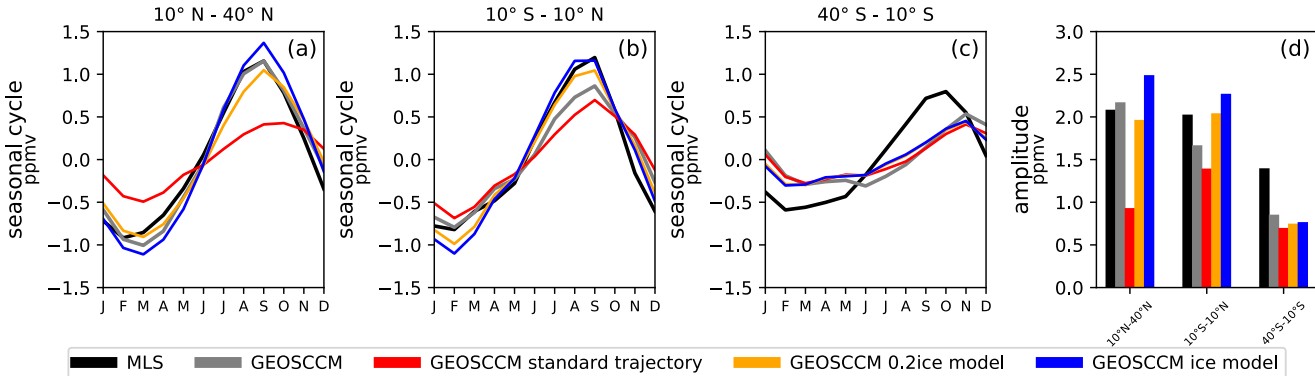

**Figure 6. Seasonal cycles of water vapor at 100 hPa averaged between (a) 10°N-40°N, (b) 10°S-10°N, and (c) 40°S-10°S and their**
**(d) seasonal amplitudes from GEOSCCM, GEOSCCM standard model, GEOSCCM ice model, and GEOSCCM 0.2ice model. We**
**have subtracted the annual mean from each data set.**

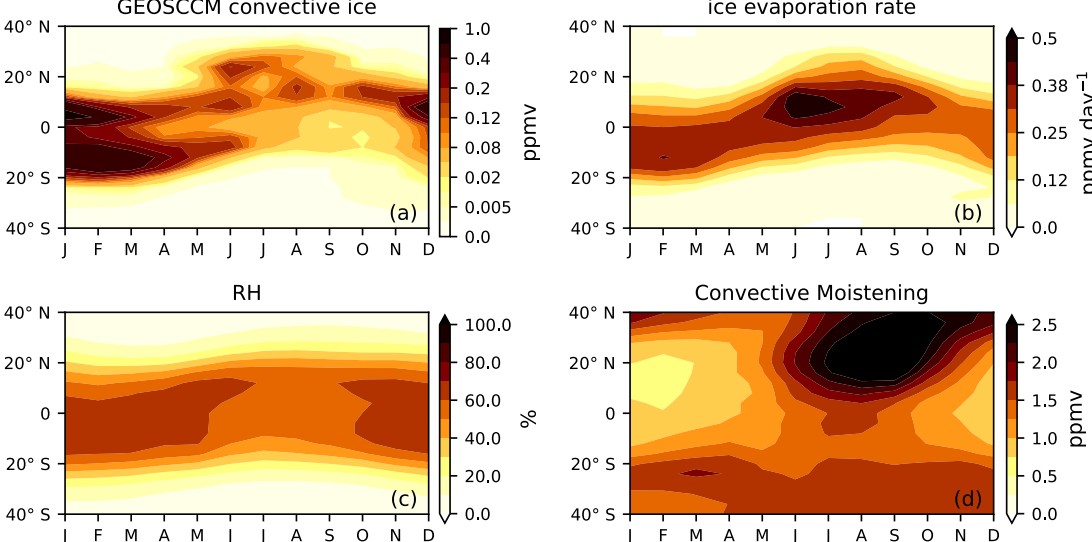

**Figure 7. (a) Zonal mean seasonal cycle of 100-hPa convective ice (ppmv) from GEOSCCM. Note the color scale is nonlinear. (b)**
**Zonal mean seasonal cycle of 100-hPa evaporation rate (ppmv day-1) from the GEOSCCM ice model. (c) Zonal mean seasonal**
**cycle of relative humidity (%) with respect to ice at 100 hPa from GEOSCCM. (d) Zonal mean seasonal cycle of net convective**
**moistening (ppmv) at 100 hPa from the ice model. The quantity net convective moistening is the difference between water vapor**
**values from the ice model and standard model.**

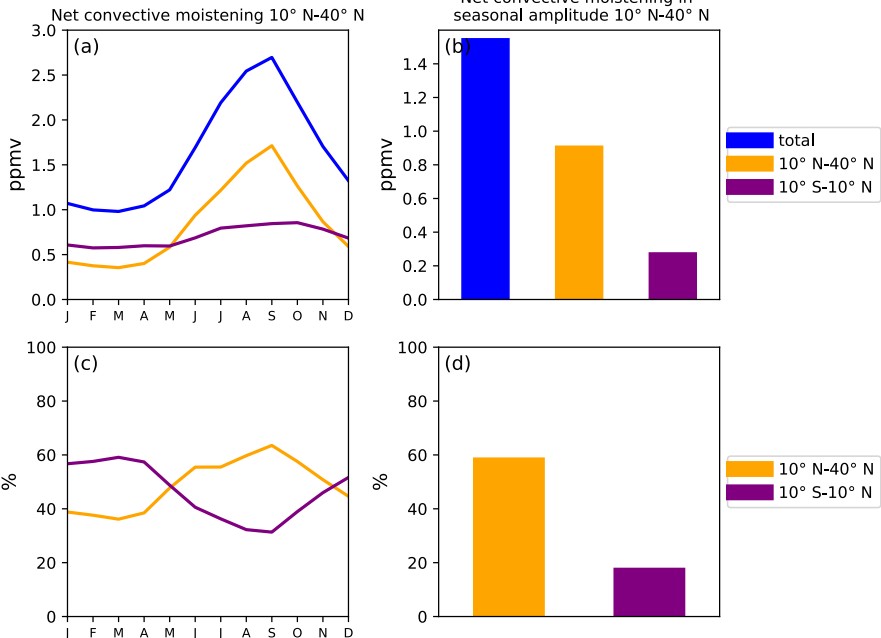

**Figure 8. (a) Net convective moistening (ppmv) in the 100-hPa 10°N-40°N water vapor seasonal cycle and the portions (ppmv)**
**contributed by convective ice evaporation over 10°S-10°N and 10°N-40°N. (b) Net convective moistening (ppmv) in the 100-hPa**
**10°N-40°N water vapor seasonal amplitude and the portions (ppmv) contributed by convective ice evaporation over 10°S-10°N and**
**10°N-40°N. (c)-(d) Percentage of net convective moistening in the 100-hPa 10°N-40°N water vapor seasonal cycle and seasonal**
**amplitude contributed by convective ice evaporation over 10°S-10°N and 10°N-40°N. The percentage is net convective moistening**
**contributed by 10°S-10°N or 10°N-40°N region divided by the total net convective moistening.**

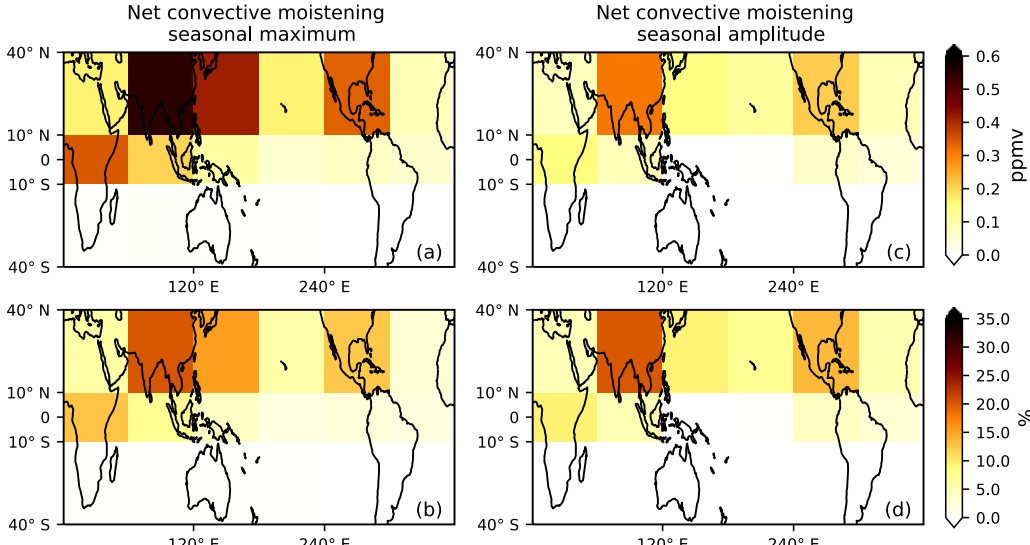

**Figure 9. Portions of net convective moistening (ppmv) in the (a) maximum value and (b) seasonal amplitude of the 100-hPa 10°N-**
**40°N water vapor seasonal cycle contributed by 12 equal-area box regions between 10°S-40°N. (c) and (d): Same as (a) and (b), but**
**for the percentage of net convective moistening contributed by the 12 equal-area box regions.**