# Peer review of "Impact of convectively lofted ice on the seasonal cycle of water vapor in the tropical tropopause layer"

_Atmospheric Chemistry and Physics, 2019_

## Short Comment (SC1) · 20 May 2019

This manuscript describes a modeling study of the impacts of deep convection on lower stratospheric water vapor and its seasonal cycle. Simple trajectory calculations are combined with the anvil ice water content (IWC) from the Goddard Earth Observing System Chemistry Climate Model (GEOSCCM). The calculated impact of convectively-generated ice on stratospheric humidity is far larger than those from recent studies using observed convective cloud-top heights (*Schoeberl et al.*, 2018; *Ueyama et al.*, 2018). In some places, the manuscript seems to indicate that the purpose of the paper is to diagnose what is happening in the GEOSCCM model. However, in the abstract and several places in the main text, the authors seem to be arguing that the modeling framework here is useful for understanding what is happening in the real atmosphere.

This distinction should be made clear such that readers are not given a misleading impression. The applicability of the modeling approach to physical processes in the real atmosphere depends on whether the GEOSCCM anvil ice product accurately represents the distribution of convectively-generated ice with respect to the tropopause. As described in detail below, I am not convinced that the realism of the GEOSCCM anvil ice water content distribution is adequately confirmed by the comparisons with measurements presented in the manuscript. Further, the conclusions presented here contradict various other lines of evidence suggesting that direct ice injection by deep convection has a relatively small impact on stratospheric humidity; acknowledgment and discussion of these discrepancies is generally lacking in the manuscript.

Figures 3 and 4 present comparisons of the anvil IWC from GEOSCCM and the total cloud IWC from CALIOP measurements. The height distributions shown in Figure 3 show a reasonable agreement. However, as noted in passing by the authors, the CALIOP cloud products include convectively-generated clouds as well as clouds formed in situ in the upper troposphere. The CloudSat cloud classification product (specifically deep convection) would be more appropriate for this comparison. Previous analyses of satellite measurements have shown that the occurrence of deep convection cloud tops drops off rapidly above about 15 km in the tropics (*Liu and Zipser*, 2005; *Massie et al.*, 2010; *Ueyama et al.*, 2018), and most clouds occurring in the tropical tropopause region are formed in situ. As a result, the CALIOP IWC (including all clouds) should be much higher than the GEOSCCM anvil IWC in the uppermost tropical troposphere; however, the comparison shown in Figure 2 for the tropical tropopause region indicates larger IWC in GEOSCCM than in the CALIOP data. This comparison suggests the GEOSCCM model had too much ice near (or above) the tropopause, which is presumably why the model predicts such a large impact of the convective ice on stratospheric humidity.

Figure 4 shows geographic distributions of the CALIOP and GEOSCCM IWC integrated between 177 and 68 hPa. This layer average is dominated by ice at the lowest model

level included (177 hPa), and it is therefore not useful for assessing the realism of the GEOSCCM anvil IWC in the vicinity of the tropical tropopause. Comparisons of the CALIOP and GEOSCCM IWC at 100 and 82 hPa would be more useful for this purpose, but again the CALIOP cloud product near the tropical tropopause is likely dominated by clouds formed in situ.

Various lines of evidence indicate that direct injection of ice into the lower stratosphere by deep convection has a relatively weak impact on stratospheric humidity. Numerous studies over the past 20 years have documented the strong correlation between tropical cold-point tropopause temperature and the lower stratospheric humidity (e.g. *Randel et al.*, 2004; *Fujiwara et al.*, 2010; *Liang et al.*, 2011; *Fueglistaler et al.*, 2013). This strong coupling would break down if direct convective injection significantly contributed to the stratospheric water vapor budget. Further, as shown by *Dessler et al.* (2007) and others, a significant contribution from sublimation of convectively-lofted ice to the lower stratospheric humidity would result in higher water isotope (HDO) enrichment than indicated by satellite and in situ observations. The current version of the manuscript does not include discussion of either of these issues.

Direct calculations conducted by a co-author on this paper indicate a far smaller impact of convective hydration on lower stratospheric humidity (*Schoeberl et al.*, 2018). The *Schoeberl et al.* (2018) study used an observation-based convective cloud-top product along with a forward-trajectory approach similar to the approach used in the current paper. With the observation-based convective cloud product, *Schoeberl et al.* (2018) concluded that convective hydration produces a less than 2% increase in stratospheric humidity. The current study using GEOSCCM anvil ice gives a gives a convective impact nearly two orders of magnitude higher (1.9 ppmv enhancement in the northern subtropical lower stratosphere). The authors should acknowledge this glaring discrepancy clearly and discuss the reasons for the difference.

The authors conclude that most of the convective moistening in their simulations comes from the Asian monsoon region. However, analyses of convective moistening using

aircraft and satellite (MLS) measurements suggest that this process primarily occurs over the north American monsoon region where the tropopause is relatively low and deep convection extends well into the lower stratosphere (*Schwartz et al.*, 2013; *Smith et al.*, 2017). Furthermore, as noted above, significant sublimation of convectively-lofted ice in the lower stratosphere would cause a large enhancement in the HDO/$H_2O$ ratio. HDO observations from the Atmospheric Chemistry Experiment (ACE) satellite show that the HDO/$H_2O$ ratio is significantly enhanced in the lowermost stratosphere over the north American monsoon region, but there is no indication of enhancement over the Asian monsoon region (*Randel et al.*, 2012). The observed HDO/$H_2O$ isotope regional distribution is inconsistent with the conclusion of this paper that convective hydration primarily occurs over the Asian monsoon.

Returning to the purpose of the manuscript, it is worth noting again that the results of the paper depend critically on the geographic distribution and height distribution of the GEOSCCM anvil ice water content product. Convective cloud in the global model is produced by a convective parameterization with poorly constrained tuning parameters, and the IWC in the model also depends on the assumed detrained ice crystal size and corresponding fall speed, which are poorly constrained by observations. As described above, the comparisons with CALIOP total cloud products are not appropriate for this purpose, and even these apples-to-oranges comparisons indicate the model has more ice mass near (or above) the tropopause than indicated by the measurements. The current version of the manuscript gives the misleading impression that the modeling approach used provides a quantitatively useful assessment of the impact of convectively-lofted ice on stratospheric humidity in the real atmosphere.

The analysis presented here might be useful as a diagnostic for understanding the control of stratospheric humidity in the GEOSCCM model. However, my understanding is that the GEOSCCM model fields used here come from a version of the model that is several years old. Newer versions of the GEOSCCM model have made substantial changes to the treatments of deep convection and stratiform ice. Recent versions of

the model anvil ice product may well have less ice mass near or above the tropopause than the version used here, in which case the analysis presented here may be of little interest to the GEOSCCM developers and users.

**Additional technical comments**

The FDF model parcels are launched at 370 K potential temperature. Particularly during Boreal summertime, the tropical cold point tropopause is often below 370 K; therefore, some of the parcels are not experiencing the true Lagrangian dry point. This error will affect both the seasonal cycle and the impact of convective hydration. The authors should re-run the simulations with parcels launched at a lower altitude (perhaps 360 K) to ensure proper representation of the tropopause cold-trap dehydration.

The manuscript states that ice forms at 80% relative humidity with respect to ice. This assumption makes some physical sense in the framework of a global Eulerian model where the cloud only occupies a fraction of the grid box. In the Lagrangian parcel model framework, this assumption is obviously unrealistic and should give a stratospheric humidity 20% too low in the absence of convective ice input. This assumption also exaggerates the impact of convective ice sublimation. Again, the 80% RH threshold could only be reasonable if the sole purpose of the paper was diagnosing processes in the GEOSCCM model.

**References**

Dessler, A. E., T. F. Hanisco, and S. Füglistaler (2007), Effects of convective ice loft-ing on H$_2$O and HDO in the tropical tropopause layer, *J. Geophys. Res.*, *112*, doi:10.1029/2007JD008,609.

Fueglistaler, S., Y. S. Liu, T. J. Flannaghan, P. H. Haynes, D. P. Dee, W. J. Read, E. E. Rems-berg, L. W. Thomason, D. F. Hurst, J. R. Lanzante, and P. F. Bernath (2013), The relation between atmospheric humidity and temperature trends for stratospheric water, *J. Geophys. Res.*, *118*, doi:10.1002/jgrd.50,157.

Fujiwara, M., H. Vömel, F. Hasebe, M. Shiotani, S.-Y. Ogino, S. Iwasaki, N. Nishi, T. Shibata, K. Shimizu, E. Nishimoto, J. M. V. Canossa, H. B. Selkirk, and S. J. Oltmans (2010), Seasonal to decadal variations of water vapor in the tropical lower stratosphere observed with balloon-borne cryogenic frost point hygrometers, *J. Geophys. Res.*, *115*, doi:10.1029/2010JD014,179.

Liang, C. K., A. Eldering, A. Gettelman, B. Tian, S. Wong, E. J. Fetzer, and K. N. Liou (2011), Record of tropical interannual variability of temperature and water vapor from a combined AIRS-MLS data set, *J. Geophys. Res.*, *116*, doi:10.1029/2010JD014,841.

Liu, C., and E. J. Zipser (2005), Global distribution of convection penetrating the tropical tropopause, *J. Geophys. Res.*, *110*, doi:10.1029/2005JD006,063.

Massie, S., J. Gille, C. Craig, R. Khosravi, J. Barnett, W. Read, and D. Winker (2010), HIRDLS and CALIPSO observations of tropical cirrus, *J. Geophys. Res.*, *115*, doi:10.1029/2009JD012,100.

Randel, W. J., F. Wu, S. J. Oltmans, K. Rosenlof, and G. E. Nedoluha (2004), Interannual changes of stratospheric water vapor and correlations with tropical tropopause temperatures, *J. Atmos. Sci.*, *99*, in press.

Randel, W. J., E. Moyer, M. Park, E. J. Jensen, P. Bernath, K. A. Walker, and C. Boone (2012), Global variations of HDO and HDO/$H_2O$ ratios in the upper troposphere and lower stratosphere derived from ace-fts satellite measurements, *J. Geophys. Res.*, *117*, doi:10.1029/2011JD016,632.

Schoeberl, M. R., E. J. Jensen, L. Pfister, R. Ueyama, M. Avery, and A. E. Dessler (2018), Convective hydration of the upper troposphere and lower stratosphere, *J. Geophys. Res.*, *123*, doi:10.1029/2018JD0282,865.

Schwartz, M. J., W. G. Read, M. L. Santee, N. J. Livesey, L. Froidevaux, A. Lambert, and G. L. Manney (2013), Convectively injected water vapor in the north American summer lowermost stratosphere, *Geophys. Res. Lett.*, *40*, doi:10.1002/grl.50,421.

Smith, J. B., D. M. Wilmouth, K. M. Bedka, K. P. Bowman, C. R. Homeyer, J. A. Dykema, M. R. Sargent, C. E. Clapp, S. S. Leroy, D. S. Sayres, J. M. Dean-Day, T. P. Bui, and J. G. Anderson (2017), A case study of convectively sourced water vapor observed in the overworld stratosphere over the united states, *J. Geophys. Res.*, *122*, doi:10.1002/2017JD026,831.

Ueyama, R., E. J. Jensen, and L. Pfister (2018), Convective influence on the humidity and clouds in the tropical tropopause layer during boreal summer, *J. Geophys. Res.*, *123*, doi:10.1029/2018JD028,674.

---

## Referee Comment (RC1) · Stephan Fueglistaler (Referee) · 5 Jun 2019

The processes that control the water budget of the TTL, and ultimately the stratosphere, remain remarkably uncertain despite much progress in observational data and numerical modeling. In particular, the moistening from ice detrained from deep convection remains poorly quantified. Wang et al. address the problem with a trajectory model, and compare the results to observations. If I understand the paper correctly, the logic is as follows: The GEOSCCM model reproduces the Microwave Limb Sounder observations in particular also during the boreal summer months well. On the other hand, the trajectory model that employs only advection and condensation (with instantaneous removal of condensate) is biased compared to the GEOSCCM water vapor field, and hence is also biased compared to MLS. Similarly, if the model is driven with renalysis

data, the patterns look similar to those obtained when driven with GEOSCCM. Applying a simple cloud model along the trajectories does not really improve the spatial pattern - the model simply increases water vapor everywhere, rather than fixing the spatial biases. I assume the cloud model is similar to that of Fueglistaler and Baker (2007, Atmos.Chem. Phys.), and as such results should be treated with caution since such a model can only serve as explorative tool - there are many poorly constrained parameters and showing results for only one configuration (which must have been tuned towards an unspecified target) may cause some concern. However, having worked with such models myself, I concur with the authors that such a model will change water vapor mixing ratios evenly, but not amplify spatial pattersn. (Figure 5b from Fueglistaler et al. (2013, J. Geophys. Res.) may be of interest to the authors, as it addresses the same problem with a similar model.) Thus, the case for convective moistening made by Wang et al. based on the spatial pattern (e.g. their Figures 1, 2 and 5) is quite convincing. However, I am very concerned whether their results are not unduly influenced - or even artefact - of the way they initialize their trajectory model. According to the paper:

"... and 60deg latitude, with initial water vapor mixing ratio of 200 parts per million by volume (ppmv). The initialization level is at 370 K potential temperature, which is above the average level of zero heating ($\sim$355-360 K) (Fueglistaler et al., 2009) but below the tropical tropopause ($\sim$380 K)."

If the initiation level is indeed 370K (and this is not a typo), then the model is essentially initialised with the driest possible state in the colder regions (over the cold Western Pacific, the tropopause is around 370K), and it is trivial that the model has a dry bias below that level - which is a problem particularly for the 100hPa level. (I should add that the display of data on pressure levels - necessary for MSL - makes it very difficult for the reader to understand whether the model is initialized above or below that pressure level.) Abover 370K, this model may have trajectories that never experienced a true cold point, but the biggest problem is below that level: in this model, anything below

370K is populated only by descending pathways - and none have just directly come up from the troposphere (which can be expected to be at local saturation, and hence much moister). This most certainly biases the results strongly. Further - are the authors really initializing the trajectories in the extratropical stratosphere up to 60 degrees latitude with 200ppmv as stated in the text? This seems dramatically off - I am puzzled how this would produce the (reasonably looking) patterns shown in Figure 5?

The serious concerns with the initialization preclude further discussion of the results since the quantitative numbers for convective moistening may be strongly affected. This paper tackles an important problem, but the authors need to demonstrate that their results are not unduly influenced by the initialization - which should be done in the troposphere and not at the tropopause. Major revisions are therefore inevitable, unfortunately.

――――――――――――――――――――

---

## Referee Comment (RC2) · William Read (Referee) · 10 Jul 2019

This paper builds on much prior work performed by some of the authors that is attempting to quantify the role convection plays on the water vapor budget in the stratosphere. The paper comes up with the basic conclusion that convectively lofted ice adds only a minor amount of water between 40S-10N (ie tropical tropopause temperature control can explain the measurements and detailed climate model runs) but is necessary between 10N-40N. This is not necessarily a new finding as prior references have shown. The new findings here is to use a trajectory model with a convective ice component to diagnose and quantify the contribution of convective ice to the fields and seasonal cycle of H2O in the northern subtropics. I beleive the authors do a good job with this and I am satisfied that this publication is good to go.

[Figure]

Just a minor query:

Since this is mostly diagnosing an output result from the GEOSCCM I dont see what the pupose of the model runs done with ERAi and MERRA-2 add to this. I think figures 1 and 2 could have been made from just using trajectories driven from GEOSCCM thus simplifying the analysis and figures. I assume that driving the trajectory model with any of these wind and temperature fields will produce similar results. Ie compare figure 5 to 1 and 2. Maybe I just missed something here.

On page 11 line 325 it –> in.

---

## Author Comment (AC1) · 15 Aug 2019

1. "Above 370K, this model may have trajectories that never experienced a true cold point, but the biggest problem is below that level: in this model, anything below 370K is populated only by descending pathways - and none have just directly come up from the troposphere (which can be expected to be at local saturation, and hence much moister). This most certainly biases the results strongly... This paper tackles an important problem, but the authors need to demonstrate that their results are not unduly influenced by the initialization - which should be done in the troposphere and not at the tropopause..."

This is a good point. We have re-run the model with parcels initialized at 360 K. While

some quantitative details of the plots have changed, the trajectory model is still unable to simulate the seasonal cycle in the NH subtropics when water vapor is influenced only by temperature.

Fig. 1 shows the trajectory models initialized at 360 K. It still shows smaller seasonal oscillation in the NH subtropics and smaller north-south asymmetry in the seasonal cycle compared to the MLS data. The figure also shows that adding the cloud model (Schoeberl et al., 2014) does not improve agreement.

We also did three test runs using ERAi, MERRA2, and GEOSCCM meteorology where we initialized the parcels above the local level of zero heating rate but not above the tropopause (Fig. 2). These runs agree closely with those initialized at 360 K.

In the revised version of the paper, we will replace all of the runs with those that initialize the parcels at 360 K.

Minor comments:

The reviewer asked: "Further - are the authors really initializing the trajectories in the extratropical stratosphere up to 60 degrees latitude with 200 ppmv as stated in the text?" The answer is, yes, we initialize parcels well into mid-latitudes. The reason these parcels do not impact the 100-hPa water values is that most of these mid- and high-latitude parcels are descending, so they never reach 100 hPa. Any that do ascend are immediately dehydrated by cold temperatures, so their water vapor values are set by TTL temperatures by the time they get to 100 hPa. We will add a statement to the paper discussing this.

Reference

Schoeberl, M. R., Dessler, A. E., Wang, T., Avery, M. A. and Jensen, E. J.: Cloud formation, convection, and stratospheric dehydration, Earth Sp. Sci., 1(1), 1–17, doi:10.1002/2014EA000014, 2014.

[Figure]

[Figure]

**Fig. 1.** Zonal mean water vapor seasonal cycle at 100, 82, and 68 hPa from MLS and trajectory models. The trajectory models are initialized at 360 K. We used the averaging kernels for all modeled water vapor.

ERAi trajectory above ZH     MERRA2 trajectory above ZH     GEOSCCM trajectory above ZH

**Fig. 2.** Zonal mean water vapor seasonal cycle at 100, 82, and 68 hPa from ERAi, MERRA2, and GEOSCCM trajectory models. The trajectory models are initialized at local levels above zero heating rate.

---

## Author Comment (AC3) · 17 Aug 2019

1. "In some places, the manuscript seems to indicate that the purpose of the paper is to diagnose what is happening in the GEOSCCM model. However, in the abstract and several places in the main text, the authors seem to be arguing that the modeling framework here is useful for understanding what is happening in the real atmosphere. This distinction should be made clear such that readers are not given a misleading impression."

Our goal is to use the GEOSCCM to help us interpret the MLS data. The argument we are making is this:

1) We have identified features in the MLS data that cannot be reproduced by trajectory models just using temperature to regulate water vapor: The MLS shows that the seasonal oscillation of 100-hPa water vapor in the northern hemispheric subtropics is larger than that in the southern hemispheric subtropics. A trajectory model driven by temperature and transport doesn't reproduce this. Adding more detailed microphysics (the Cloud Model) to the dehydration process of the trajectory model doesn't reproduce the observed hemispheric asymmetry either.

2) We show that a chemistry-climate model reproduces the MLS water vapor seasonal cycle.

3) In that model, we show that moistening by convective ice evaporation is responsible for the hemispheric asymmetry in the water vapor seasonal cycle.

4) We argue that this gives us insight into what's going on in the real world. Obviously, the amount of credence someone gives this argument is a judgment call. But it is our view that this argument is quite strong, particularly since there is no competing hypothesis for the asymmetry. And neither of the other reviewers had a problem with our approach.

In the revised version, we will edit the document to make sure the chain of logic in the paper is clear.

2. "The height distributions shown in Figure 3 show a reasonable agreement. However, as noted in passing by the authors, the CALIOP cloud products include convectively-generated clouds as well as clouds formed in situ in the upper troposphere..."

"Figure 4 shows geographic distributions of the CALIOP and GEOSCCM IWC integrated between 177 and 68 hPa. This layer average is dominated by ice at the lowest model level included (177 hPa), and it is therefore not useful for assessing the realism of the GEOSCCM anvil IWC in the vicinity of the tropical tropopause. Comparisons of the CALIOP and GEOSCCM IWC at 100 and 82 hPa would be more useful for this purpose.."

This is a good point and one that requires us to make clear a few points. First, our goal is not to validate the GEOSCCM; we agree that our comparisons to CALIOP suggests it has too much convective ice in the TTL. Rather, as we described above, our goal is to show that differences between the MLS and temperature-only trajectory models also exist in a parallel analysis of the GEOSCCM; and that the differences in the GEOSCCM analysis can be largely fixed if we add convection into the trajectory model. As described below, we have performed sensitivity analyses that show that this conclusion is robust to the amount of convective ice in the GEOSCCM, so convective ice overestimates in the GEOSCCM do not impact our result.

To make these points clear, we will substantially edit our discussion of GEOSCCM convective ice. We show in Figure 1 an updated comparison between CALIOP and GEOSCCM ice data. For the CALIOP, we show the ice from all clouds minus the ice from thin cirrus clouds above 146 hPa during 200805 – 201312 (private communication from Tao Wang), which is a rough estimate of convective ice in the TTL region, although it is almost certainly an underestimate of true convective ice amount. For the GEOSCCM, we show the total convective ice, as well as the convective ice decreased by 90% and 80%. The vertical profile of 30N-30S convective ice (Fig. 2) shows that a decrease of 80% brings tropical mean GEOSCCM convective ice into better agreement with CALIOP values at 121 hPa and above.

Fig. 3 shows the distribution of convective ice averaged between 121 – 82 hPa during JJA (top) and DJF (bottom). We then use the decreased convective ice data from the GEOSCCM to produce two trajectory test runs. Fig. 4 shows the 100-hPa water vapor seasonal cycle produced by the GEOSCCM and by trajectory models with convective ice decreased by various amounts as well as one run with no ice. Reducing the GEOSCCM IWC to bring it into agreement with CALIOP IWC does not change the conclusion that adding ice helps improve the seasonal cycle in the trajectory simulation of the GEOSCCM.

3. "Further, the conclusions presented here contradict various other lines of evidence

suggesting that direct ice injection by deep convection has a relatively small impact on stratospheric humidity; acknowledgment and discussion of these discrepancies is generally lacking in the manuscript."

"Various lines of evidence indicate that direct injection of ice into the lower stratosphere by deep convection has a relatively weak impact on stratospheric humidity. Numerous studies over the past 20 years have documented the strong correlation between tropical cold-point tropopause temperature and the lower stratospheric humidity (e.g. Randel et al., 2004; Fujiwara et al., 2010; Liang et al., 2011; Fueglistaler et al., 2013). This strong coupling would break down if direct convective injection significantly contributed to the stratospheric water vapor budget."

"Direct calculations conducted by a co-author on this paper indicate a far smaller impact of convective hydration on lower stratospheric humidity (Schoeberl et al., 2018)..."

Previous studies, including those that Jensen cites, show strong coupling between the fluctuations in water vapor and fluctuations in the TTL temperature on interannual time scales. We agree that these are tightly connected - in fact, Dessler et al. (2016) showed that the GEOSCCM also reproduces this tight coupling (see Fig. 4 of that paper and discussion). However, none of these papers quantify the impact of convection on the seasonal cycle and therefore do not contradict our analysis.

Schoeberl et al. (2018) quantifies convection and concludes it is not important, but his analysis only covered DJF of 2008/09. Thus, that paper does not tell us anything about the main conclusions of our paper - that summertime convection is important for NH seasonal cycle. Fueglistaler et al. (2013), pointed out that the trajectory model using the Lagrangian Dry Point (LDP) has a dry bias when predicting the annual mean entry water vapor. They then showed that employing a cloud microphysical box model for the dehydration better reproduces the observed annual mean entry water vapor in the tropics ($25°$N-$25°$S), but tends to underestimate the seasonal amplitude of the MLS and HALOE water vapor. Their result is consistent with the conclusion of this paper

that temperature variations alone are not sufficient to produce all the features of the LS water vapor seasonal cycle.

4. "Further, as shown by Dessler et al. (2007) and others, a significant contribution from sublimation of convectively-lofted ice to the lower stratospheric humidity would result in higher water isotope (HDO) enrichment than indicated by satellite and in situ observations."

We agree that more work should be done on HDO and we have now put in a statement to that effect. However, we do not think that this is a strong argument against our conclusions. One thing that is clear in the 20+ years people have been analyzing stratospheric HDO is that it is not a strong constraint on water vapor processes and multiple sets of processes can produce the observed HDO fields. For example, Dessler et al. (2007) argues convection is required to explain stratospheric HDO, but Gettelman and Webster (2005) argue that it is not required. It is our hope that our paper will motivate future work on this issue.

5. "The authors conclude that most of the convective moistening in their simulations comes from the Asian monsoon region. However, analyses of convective moistening using aircraft and satellite (MLS) measurements suggest that this process primarily occurs over the north American monsoon region where the tropopause is relatively low and deep convection extends well into the lower stratosphere (Schwartz et al., 2013; Smith et al., 2017)."

Neither of the papers cited actually say what Jensen says they do. Schwartz et al. (2013) shows that extremely large water vapor values are observed by the MLS over both North America (NA) and the Asian monsoon anticyclone region (AMA). They make no claims about whether the NA monsoon is more or less important than the Asian monsoon. The study by Smith et al. (2017) shows that deep convection is observed to contribute to high frequency of enhanced water vapor over the central U.S. during boreal summer. Their paper doesn't reach any conclusions about the Asian monsoon

region. Thus, neither of these papers contradict anything in our paper.

It is possible that the GEOSCCM trajectory model underestimates the convective impact over the NA region, since at 100 and 82 hPa, the GEOSCCM underestimates the convective ice amount in the NA region. This is mentioned in the submitted paper (line 281-284). In the revised paper, we'll make this point clearer.

6. "The FDF model parcels are launched at 370 K potential temperature. Particularly during Boreal summertime, the tropical cold point tropopause is often below 370 K; therefore, some of the parcels are not experiencing the true Lagrangian dry point..."

This is a good point and we have addressed it by re-running the model simulations with parcels initialized at 360 K. The analysis and plots in the paper will be updated with these new runs. While some details of the plots have changed, the trajectory model is still unable to simulate the seasonal cycle in the NH subtropics when water vapor is regulated only by temperature (Fig. 5).

We also did three test runs using ERAi, MERRA2, and GEOSCCM meteorology where we initialize the parcels above the local level of zero heating rate (Fig. 6 below). These runs agree closely with those initialized at 360 K and they also show that temperature alone can't reproduce the larger LS water vapor seasonal cycle in the NH subtropics as shown in the MLS and GEOSCCM.

We will replace all analyses in the revised version of the paper with runs initialized at 360 K.

7. "The manuscript states that ice forms at 80% relative humidity with respect to ice..."

It is correct that we limit water vapor in the parcels to 80% relative humidity. We could've used 100% relative humidity and the conclusion about the hemispheric asymmetry would remain the same, except the water vapor values would be higher everywhere by a factor of 1.1-1.2 at 100 hPa and 82 hPa. The choice of saturation threshold does not disproportionately affect ice evaporation. As discussed in Dessler et al., (2016) in

some detail, our choice of 80% is motivated by the fact that the trajectory model fields agree better with the GEOSCCM than when using 100%.

The key point here is that, whether using 100% or 80% threshold, convective moistening still needs to be included in order to reproduce the seasonal cycle in the NH subtropics and the hemispheric asymmetry. To demonstrate this, we show the zonal mean water vapor seasonal cycles from the GEOSCCM trajectory model using the 100% RH saturation threshold together with that using the 80% RH saturation threshold (Fig. 7). Indeed, the water vapor values in the 100%-RH run are about 1.1 to 1.2 larger everywhere at 100 hPa and 82 hPa compared to the 80%-RH run. However, this does not reproduce the observed seasonal cycle in the NH subtropics or the hemispheric asymmetry. Thus, our results are robust to the choice of threshold.

References

Dessler, A. E., Hanisco, T. F. and Fueglistaler, S.: Effects of convective ice lofting on H 2 O and HDO in the tropical tropopause layer, J. Geophys. Res., 112(D18), D18309, doi:10.1029/2007JD008609, 2007.

Dessler, A. E., Ye, H., Wang, T., Schoeberl, M. R., Oman, L. D., Douglass, A. R., Butler, A. H., Rosenlof, K. H., Davis, S. M. and Portmann, R. W.: Transport of ice into the stratosphere and the humidification of the stratosphere over the 21st century, Geophys. Res. Lett., 43(5), 2323–2329, doi:10.1002/2016GL067991, 2016.

Fueglistaler, S., Liu, Y. S., Flannaghan, T. J., Haynes, P. H., Dee, D. P., Read, W. J., Remsberg, E. E., Thomason, L. W., Hurst, D. F., Lanzante, J. R. and Bernath, P. F.: The relation between atmospheric humidity and temperature trends for stratospheric water, J. Geophys. Res. Atmos., 118(2), 1052–1074, doi:10.1002/jgrd.50157, 2013.

Gettelman, Andrew; Webster, C. R.: Simulations of water isotope abundances in the upper troposphere and lower stratosphere and implications for stratosphere troposphere exchange, J. Geophys. Res., 110(D17), D17301, doi:10.1029/2004JD004812,

2005.

Schoeberl, M. R., Jensen, E. J., Pfister, L., Ueyama, R., Avery, M. and Dessler, A. E.: Convective Hydration of the Upper Troposphere and Lower Stratosphere, J. Geophys. Res. Atmos., 123(9), 4583–4593, doi:10.1029/2018JD028286, 2018.

Schwartz, M. J., Read, W. G., Santee, M. L., Livesey, N. J., Froidevaux, L., Lambert, A. and Manney, G. L.: Convectively injected water vapor in the North American summer lowermost stratosphere, Geophys. Res. Lett., 40(10), 2316–2321, doi:10.1002/grl.50421, 2013.

Smith, J. B., Wilmouth, D. M., Bedka, K. M., Bowman, K. P., Homeyer, C. R., Dykema, J. A., Sargent, M. R., Clapp, C. E., Leroy, S. S., Sayres, D. S., Dean-Day, J. M., Paul Bui, T. and Anderson, J. G.: A case study of convectively sourced water vapor observed in the overworld stratosphere over the United States, J. Geophys. Res. Atmos., 122(17), 9529–9554, doi:10.1002/2017JD026831, 2017.

[Figure]

**Fig. 1.** CALIOP, GEOSCCM, and reduced GEOSCCM zonal mean convective ice in a pressure-latitude domain.

[Figure]

**Fig. 2.** CALIOP, GEOSCCM, and reduced GEOSCCM convective ice (ppmv) profile averaged between 30°N- 30°S.

[Figure]

**Fig. 3.** Same as Fig. 1, but for CALIOP and GEOSCCM ice averaged between 121 – 82 hPa in a latitude- longitude domain.

[Figure]

**Fig. 4.** Zonal mean water vapor seasonal cycle at 100 hPa from GEOSCCM, GEOSCCM trajectory, and GEOSCCM trajectory with with convective ice moistening. The trajectory models are initialized at 360 K.

[Figure]

**Fig. 5.** Zonal mean water vapor seasonal cycle at 100, 82, and 68 hPa from MLS and trajectory models. The trajectory models are initialized at 360 K. We used the averaging kernels for all modeled water vapor.

**Fig. 6.** Zonal mean water vapor seasonal cycle at 100, 82, and 68 hPa from MLS and trajectory models. The trajectory models are initialized at local levels above zero heating rate.

**Fig. 7.** Zonal mean water vapor seasonal cycles at 100, 82, and 68 hPa from GEOSCCM and trajectory models. We show trajectory models initialized at 360 K, using two saturation thresholds: 80% RH and 100% RH.

---

## Short Comment (SC2) · 4 Sep 2019

[titlepage,12pt]article

natbib graphicx indentfirst times color

**Second comment on "Impact of convectively lofted ice on the seasonal cycle of tropical lower stratospheric water vapor" by X. Wang et al.**

In their response (AC3) to my first comment, the authors have partially addressed some of my concerns. However, some of the important issues were side-stepped, and some of the issues were dismissed based on inaccurate statements. Details are provided below.

1. An important issue not addressed in my previous comment is the similarity between the current analysis of convective impact on UTLS water and the *Ueyama et al.* (2015) and *Ueyama et al.* (2018) analyses. These studies used a much more detailed model with full treatment cloud microphysics and vertical redistribution of water vapor by both in situ clouds and convective clouds. A convective cloud-top product derived from observations was used. The focus of the Ueyama et al. studies was processes controlling water vapor at 100 hPa, including deep convection. They showed that convective hydration has a significant impact in both Boreal winter (2015 paper) and summer (2018 paper). They showed that convective hydration is responsible for much of the geographic structure in 100-hPa $H_2O$ during summertime observed by MLS, and the overall tropical-mean 100-hPa water vapor is increased substantially during summertime. Together, the papers showed that convective influence increases 100-hPa water vapor more during summertime than wintertime. Hence, these studies already showed that convective influence was responsible for much of the seasonal cycle in MLS-observed water vapor at 100-hPa during summertime. The *Wang et al.* manuscript does not cite or discuss the Ueyama et al. papers. This oversight should be corrected.

2. As noted in the first comment, the plausibility of the *Wang et al.* results depends entirely on how well the GEOSCCM convective ice water content product represents the occurrence of convective clouds above the tropopause. In their response to my first comment, the authors state that "our goal is not to validate the GEOSCCM." Yet, their use of multiple figures to compare the GEOSCCM convective IWC to CALIOP

observations is obviously an attempt to validate this aspect of the model. In fact, the discussion in the original manuscript gives the impression that the GEOSCCM product is perfectly reasonable.

Even more alarming is that the "convective" subset of CALIOP IWC shown in AC3 Figure 1 seems to extend to much higher altitudes than the full CALIOP IWC shown in Figure 3 of the originally submitted manuscript. In Figure 3 of the submitted manuscript, the occurrence of all clouds detected by CALIOP appears to drop off to near zero just above 100 hPa. In Figure 1 of AC3, the "convective" subset extends to above 68 hPa! This contrast is physically unrealistic, and it is not clear what the authors are actually doing with the data. Are they really claiming that convective clouds extend above 68 hPa as indicated in Fig. 1 of AC3?

Lastly, and perhaps most importantly, the critical issue here is how much IWC the GEOSCCM product predicts to be above the *local* tropopause. The basic result of the paper is that sublimation of convectively-lofted ice in the lower stratosphere is an important source term in the stratospheric water vapor budget. This result depends entirely on how much ice exists above the local tropopause where it will sublimate in the warm, dry lower stratosphere. None of the figures shown by *Wang et al.* directly evaluates the plausibility of GEOSCCM convective ice occurrence above the local tropopause. Addressing this issue would require examination of the GEOSCCM IWC field in tropopause-relative coordinates. This evaluation is critical in order to determine the plausibility of the central results of the paper.

3. In AC3, the authors dismiss the value of water vapor isotope measurements for assessment of transport and hydration/dehydration mechanisms. In fact, as noted extensively in the literature, the HDO/$H_2O$ fraction is extremely sensitive to sublimation of convectively-lofted ice. If nothing else, water isotope measurements are very useful for identifying where this process might be occurring and contributing significantly to the water vapor budget. *Randel et al.* (2012) used global ACE-FTS data to examine the

geographic distribution of HDO/H$_2$O fraction, and they showed that HDO enrichment is clearly evident over the north American monsoon, but there is no indication of such HDO enrichment over the Asian monsoon. The ACE-FTS isotope measurements are consistent with the in situ and MLS measurements indicating convective plumes with enhanced H$_2$O deep in the stratosphere over the north American monsoon (*Schwartz et al.*, 2013; *Smith et al.*, 2017). As noted previously, the calculations presented here based on the GEOSCCM convective IWC product indicate lower-stratospheric hydration is predominantly occurring over the Asian monsoon, which is in direct conflict with available observational evidence.

In AC3, the authors state that *Schwartz et al.* (2013) shows anomalously high water vapor mixing ratios over both monsoon regions at 100 hPa. This statement is apparently meant to be interpreted as evidence that convective hydration of the lower stratosphere is occurring over both monsoons. However, as is well known, the tropopause is very high over the Asian monsoon (near 390 K potential temperature, about 82 hPa), therefore 100 hPa is well within the troposphere over the Asian monsoon. In contrast, the tropopause is relatively low over the north American monsoon region, and many of the anomalously high water vapor concentrations at 100 hPa are well within the stratosphere. Again, the observational evidence suggests convective overshooting deep into the stratosphere occurs over the north American monsoon region but not over the Asian monsoon region. The modeling results presented here indicate the exact opposite. This is an important point that should be addressed by the authors.

3. In AC3, the authors dismissed the *Schoeberl et al.* (2018) paper as irrelevant because it only addressed the convective influence on stratospheric humidity during wintertime. As a reminder, *Schoeberl et al.* (2018) used much the same model as the *Wang et al.* paper here, yet they used the observation-based convection product described in the *Ueyama et al.* papers discussed above. They showed that convective influence has a minimal impact on wintertime stratospheric humidity. This paper actually does conflict with the current study. Examination of the *Wang et al.* paper figures

indicates they are getting a substantial increase in stratospheric humidity throughout the year. The obvious difference is that *Schoeberl et al.* (2018) used the observation-based convective cloud-top product which indicates that convection extends above the local tropopause far less frequently than indicated by the GEOSCCM convective IWC product. The discrepancy between the *Wang et al.* model results and those of *Schoeberl et al.* (2018) should be addressed.

4. One last note: In AC3, the authors now show that the GEOSCCM convective IWC should be reduced by a factor of 5 (or 10) to provide better agreement with observations. If the GEOSCCM convective IWC is far too high, as the authors now seem to agree, that presumably invalidates the results of *Dessler et al.* (2016). The earlier study used the GEOSCCM convective IWC along with the trajectory model to argue that convective ice sublimation in the stratosphere accounts for a significant fraction (20–50%) of the global-model-predicted increase in stratospheric humidity over the 21st century. Presumably, if the GEOSCCM convective IWC is far too high, these estimates of convective contribution to future stratospheric humidity increase are also much too high. This issue should be acknowledged by the authors.

**References**

Dessler, A. E., T. Wang, M. R. Schoeberl, L. D. Oman, A. R. Douglass, A. H. Butler, K. H. Rosenlof, S. M. Davis, and R. W. Portmann (2016), Transport of ice into the stratosphere and the humidification of the stratosphere over the 21st century, *Geophys. Res. Lett.*, *43*, doi:10.1002/2016GL067,991.

Randel, W. J., E. Moyer, M. Park, E. J. Jensen, P. Bernath, K. A. Walker, and C. Boone (2012), Global variations of HDO and HDO/$H_2O$ ratios in the upper troposphere and lower stratosphere derived from ace-fts satellite measurements, *J. Geophys. Res.*, *117*, doi:10.1029/2011JD016,632.

Schoeberl, M. R., E. J. Jensen, L. Pfister, R. Ueyama, M. Avery, and A. E. Dessler (2018), Convective hydration of the upper troposphere and lower stratosphere, *J. Geophys. Res.*, *123*, doi:10.1029/2018JD0282,865.

Schwartz, M. J., W. G. Read, M. L. Santee, N. J. Livesey, L. Froidevaux, A. Lambert, and G. L. Manney (2013), Convectively injected water vapor in the north American summer lowermost stratosphere, *Geophys. Res. Lett.*, *40*, doi:10.1002/grl.50,421.

Smith, J. B., D. M. Wilmouth, K. M. Bedka, K. P. Bowman, C. R. Homeyer, J. A. Dykema, M. R. Sargent, C. E. Clapp, S. S. Leroy, D. S. Sayres, J. M. Dean-Day, T. P. Bui, and J. G. Anderson (2017), A case study of convectively sourced water vapor observed in the overworld stratosphere over the united states, *J. Geophys. Res.*, *122*, doi:10.1002/2017JD026,831.

Ueyama, R., E. J. Jensen, L. Pfister, and J.-E. Kim (2015), Dynamical, convective, and microphysical control on wintertime distributions of water vapor and clouds in the tropical tropopause layer, *J. Geophys. Res.*, *120*, doi:10.1002/2015JD023,318.

Ueyama, R., E. J. Jensen, and L. Pfister (2018), Convective influence on the humidity and clouds in the tropical tropopause layer during boreal summer, *J. Geophys. Res.*, *123*, doi:10.1029/2018JD028,674.

---

## Author Comment (AC4) · 21 Sep 2019

1. "An important issue not addressed in my previous comment is the similarity between the current analysis of convective impact on UTLS water and the Ueyama et al. (2015) and Ueyama et al. (2018) analyses. These studies used a much more detailed model with full treatment cloud microphysics and vertical redistribution of water vapor by both in situ clouds and convective clouds. A convective cloud-top product derived from observations was used. The focus of the Ueyama et al. studies was processes controlling water vapor at 100 hPa, including deep convection. They showed that convective hydration has a significant impact in both Boreal winter (2015 paper) and summer (2018 paper). They showed that convective hydration is responsible for much of the geo-graphic structure in 100-hPa H2O during summertime observed by

MLS, and the over-all tropical-mean 100-hPa water vapor is increased substantially during summertime. Together, the papers showed that convective influence increases 100-hPa water vapor more during summertime than wintertime. Hence, these studies already showed that convective influence was responsible for much of the seasonal cycle in MLS-observed water vapor at 100-hPa during summertime. The Wang et al. manuscript does not cite or discuss the Ueyama et al. papers. This oversight should be corrected."

We agree that we should reference Ueyama et al. (2015, 2018) and will do so in the revised version.

While our results are broadly in agreement with Ueyama et al., we have an entirely different methodology, as Jensen describes above. Jensen may feel that a paper he is a co-author on has answered all questions about this issue, but we respectfully disagree with that assessment.

2. "As noted in the first comment, the plausibility of the Wang et al. results depends entirely on how well the GEOSCCM convective ice water content product represents the occurrence of convective clouds above the tropopause. In their response to my first comment, the authors state that "our goal is not to validate the GEOSCCM." Yet, their use of multiple figures to compare the GEOSCCM convective IWC to CALIOP is obviously an attempt to validate this aspect of the model. In fact, the discussion in the original manuscript gives the impression that the GEOSCCM product is perfectly reasonable."

No. What this result depends entirely on is that the GEOSCCM water vapor reproduces the MLS water vapor. Once that is established, we can then tear into the model to determine what processes in the model are responsible. We find that convective ice evaporation is playing a key role.

We included comparisons of the model and observed IWC field because one of the reviewers of Dessler et al. (2016) (discussed below) gave us a very hard time in the

review of that paper, and we thought that it would be useful to show those comparisons. For the purposes of this analysis, it is our opinion that the GEOSCCM IWC fields are perfectly reasonable.

"Even more alarming is that the "convective" subset of CALIOP IWC shown in AC3 Figure 1 seems to extend to much higher altitudes than the full CALIOP IWC shown in Figure 3 of the originally submitted manuscript. In Figure 3 of the submitted manuscript, the occurrence of all clouds detected by CALIOP appears to drop off to near zero just above 100 hPa. In Figure 1 of AC3, the "convective" subset extends to above 68 hPa! This contrast is physically unrealistic, and it is not clear what the authors are actually doing with the data. Are they really claiming that convective clouds extend above 68hPa as indicated in Fig. 1 of AC3?"

Figure 1 below is a re-plot of Fig. 1 from AC3, using the same units and color-scale as the Fig. 3 from the originally submitted manuscript. It confirms that the GEOSCCM has too much IWC in the TTL. However, as we showed in AC3, if we lower the IWC so that it agrees better with CALIOP (i.e., GEOSCCM IWC divided by 5), the result stays the same.

Jensen points out potential issues with the GEOSCCM putting ice too high into the stratosphere. To quantify the impact of errors in the altitude distribution, we show in Fig. 2 a run where we don't allow any ice evaporation above 90 hPa. The difference between the full convective ice evaporation run (Fig. 2c) and this test run (Fig. 2d) is small between 30°S-30°N (Fig. 2e). The larger moisture difference at higher latitudes comes from the lowermost stratosphere. We conclude that the convective ice above 90 hPa has little impact on the water vapor seasonal cycle at 100 hPa. Thus, even if the GEOSCCM puts ice too high, it does not impact our analysis.

"Lastly, and perhaps most importantly, the critical issue here is how much IWC the GEOSCCM product predicts to be above the local tropopause. The basic result of the paper is that sublimation of convectively lofted ice in the lower stratosphere is an

important source term in the stratospheric water vapor budget. This result depends entirely on how much ice exists above the local tropopause where it will sublimate in the warm, dry lower stratosphere. None of the figures shown by Wang et al. directly evaluates the plausibility of GEOSCCM convective ice occurrence above the local tropopause. Addressing this issue would require examination of the GEOSCCM IWC field in tropopause-relative coordinates. This evaluation is critical in order to determine the plausibility of the central results of the paper."

Whether or not the air is above the local tropopause is an irrelevant detail. We know that dehydration is occurring above 100 hPa, so we make no claim anywhere in the paper that convective hydration at 100 hPa is important for the bulk stratosphere — as discussed below, Schoeberl et al. (2018) shows it mostly doesn't. Our analysis focuses on 100 hPa because it's where the MLS data are available, and it is a level of interest owing to its position in the mid-TTL. Thus, we do not feel that a tropopause-relative coordinate analysis is needed here.

However, we agree that we could discuss this better in the paper, and we have done so.

3. "In AC3, the authors dismiss the value of water vapor isotope measurements for assessment of transport and hydration/dehydration mechanisms. In fact, as noted extensively in the literature, the HDO/H2O fraction is extremely sensitive to sublimation of convectively lofted ice. If nothing else, water isotope measurements are very useful for identifying where this process might be occurring and contributing significantly to the water vapor budget. Randel et al. (2012) used global ACE-FTS data to examine distribution of HDO/H2O fraction, and they showed that HDO enrichment is clearly evident over the north American monsoon, but there is no indication of such HDO enrichment over the Asian monsoon. The ACE-FTS isotope measurements are consistent with the in situ and MLS measurements indicating convective plumes with enhanced H2O deep in the stratosphere over the north American monsoon (Schwartz et al., 2013; Smith et al., 2017). As noted previously, the calculations presented here based on

the GEOSCCM convective IWC product indicate lower-stratospheric hydration is predominantly occurring over the Asian monsoon, which is in direct conflict with available observational evidence."

This essentially repeats a point Jensen made in his first comment, but does not engage the substance of our response, so we repeat our response here: "We agree that more work should be done on HDO and we have now put in a statement to that effect. However, we do not think that this is a strong argument against our conclusions. One thing that is clear in the 20+ years people have been analyzing stratospheric HDO is that it is not a strong constraint on water vapor processes and multiple sets of processes can produce the observed HDO fields. For example, Dessler et al. (2007) argues convection is required to explain stratospheric HDO, but Gettelman and Webster (2005) argue that it is not required. It is our hope that our paper will motivate future work on this issue."

We believe it is also worth noting that Ueyama et al. (2018), which Jensen mentions several times in his comments and on which Jensen is a coauthor, state in the abstract: "Parcels are most frequently hydrated by deep convection in the southern sector of the Asian monsoon anticyclone and subsequently dehydrated downstream of convection to the west, shifting the locations of final dehydration northwest of the cold temperature region in the northern Tropics." Thus, our analysis agrees with Jensen's prior work.

As we said in our last response, we will edit the text in the paper to acknowledge that analyses of HDO would be beneficial.

"In AC3, the authors state that Schwartz et al. (2013) shows anomalously high water vapor mixing ratios over both monsoon regions at 100 hPa. This statement is apparently meant to be interpreted as evidence that convective hydration of the lower stratosphere is occurring over both monsoons. However, as is well known, the tropopause is very high over the Asian monsoon (near 390 K potential temperature, about 82 hPa), therefore 100 hPa is well within the troposphere over the Asian monsoon. In contrast, the tropopause is relatively low over the north American monsoon region, and many of the anomalously high water vapor concentrations at 100 hPa are well within the stratosphere. Again, the observational evidence suggests convective overshooting deep into the stratosphere occurs over the north American monsoon region but not over the Asian monsoon region. The modeling results presented here indicate the exact opposite. This is an important point that should be addressed by the authors."

We agree with this point. We do not intend to make claims that 100 hPa is in the stratosphere at all locations. We acknowledge that this point may not be sufficiently clear in the manuscript and will make it clear in the revised manuscript.

4. "In AC3, the authors dismissed the Schoeberl et al. (2018) paper as irrelevant because it only addressed the convective influence on stratospheric humidity during wintertime... They showed that convective influence has a minimal impact on wintertime stratospheric humidity. This paper actually does conflict with the current study... Examination of the Wang et al. paper figures indicates they are getting a substantial increase in stratospheric humidity throughout the year..."

The clear resolution of this discrepancy is to point out that Schoeberl et al. (2018) was looking at 18-30 km average water vapor, while our paper examines the summertime 100-hPa surface. There is dehydration occurring above 100 hPa, so Schoeberl et al. (2018) can claim correctly that there is not much of an effect in the bulk of the stratosphere and we can correctly claim that there is an observable impact at 100 hPa.

We additionally note that Schoeberl et al. (2018) used the same observation-based convection product described in the Ueyama et al. papers, but Ueyama found a much bigger response at 100 hPa. This provides additional support for our interpretation of these papers.

We will edit the text in the paper to make sure there is no confusion on this issue.

5. "One last note: In AC3, the authors now show that the GEOSCCM convective IWC

should be reduced by a factor of 5 (or 10) to provide better agreement with observations. If the GEOSCCM convective IWC is far too high, as the authors now seem to agree, that presumably invalidates the results of Dessler et al. (2016). The earlier study used the GEOSCCM convective IWC along with the trajectory model to argue that convective ice sublimation in the stratosphere accounts for a significant fraction (20–50%) of the global-model-predicted increase in stratospheric humidity over the 21st century."

The point of Dessler et al. (2016) was to diagnose the cause of the trend in the model. Dessler et al. (2016) did indeed find that convective ice evaporation was responsible for a significant part of the long-term trend in the model and that conclusion is still true, regardless of whether the model's IWC fields are accurate.

Jensen incorrectly implies that Dessler et al. claimed that the model's IWC fields were correct — in fact, Dessler et al. went out of their way to say that their analysis should encourage more research on the reality of the GEOSCCM's convective ice field. Here is a quote from that paper: "Nevertheless, the CCMs' predictions of ice lofting into the lower stratosphere have not been quantitatively tested against observations. The CCMs' predictions rely on their convective parameterizations, and until verified with observations, one could reasonably question the realism of their representation of the infrequent but intense convective systems that penetrate the stratosphere."

"Presumably, if the GEOSCCM convective IWC is far too high, these estimates of convective contribution to future stratospheric humidity increase are also much too high. This issue should be acknowledged by the authors."

This is an elementary error: Jensen mistakes a bias in the mean field with a bias in the feedback. It is well-known, for example, that many climate models have biases in their water vapor field when compared to observations. However, all the models predict the same water vapor feedback because they all predict the same increase in water vapor with temperature. Thus, a bias in the stratospheric ice, when compared to observations, does NOT mean that the change in ice water content over the 21st-

century as the climate warms is also wrong. It may be — and subsequent research may show that it is – but until then we can't say anything about the reality of the long-term trend in the model.

Because we believe Jensen's comments on this point are without merit, we have not made any changes to the manuscript in response to this.

References

Dessler, A. E., Hanisco, T. F. and Fueglistaler, S.: Effects of convective ice lofting on H 2 O and HDO in the tropical tropopause layer, J. Geophys. Res., 112(D18), D18309, doi:10.1029/2007JD008609, 2007.

Dessler, A. E., Ye, H., Wang, T., Schoeberl, M. R., Oman, L. D., Douglass, A. R., Butler, A. H., Rosenlof, K. H., Davis, S. M. and Portmann, R. W.: Transport of ice into the stratosphere and the humidification of the stratosphere over the 21st century, Geophys. Res. Lett., 43(5), 2323–2329, doi:10.1002/2016GL067991, 2016.

Gettelman, Andrew; Webster, C. R.: Simulations of water isotope abundances in the upper troposphere and lower stratosphere and implications for stratosphere troposphere exchange, J. Geophys. Res., 110(D17), D17301, doi:10.1029/2004JD004812, 2005.

Schoeberl, M. R., Jensen, E. J., Pfister, L., Ueyama, R., Avery, M. and Dessler, A. E.: Convective Hydration of the Upper Troposphere and Lower Stratosphere, J. Geophys. Res. Atmos., 123(9), 4583–4593, doi:10.1029/2018JD028286, 2018.

Ueyama, R., Jensen, E. J., Pfister, L. and Kim, J.-E.: Dynamical, convective, and microphysical control on wintertime distributions of water vapor and clouds in the tropical tropopause layer, J. Geophys. Res. Atmos., 120(19), 10,410-483,500, doi:10.1002/2015JD023318, 2015.

Ueyama, R., Jensen, E. J. and Pfister, L.: Convective Influence on the Humidity and Clouds in the Tropical Tropopause Layer During Boreal Summer, J. Geophys. Res.

Atmos., doi:10.1029/2018JD028674, 2018.

[Figure]

[Figure]

**Fig. 1.** . CALIOP, GEOSCCM, and reduced GEOSCCM zonal mean convective ice (mg m-3) in a pressurelatitude domain.

[Figure]

**Fig. 2.** Zonal mean water vapor seasonal cycle at 100 hPa from GEOSCCM and GEOSCCM trajectory models (1-d). Panel e shows the difference between panel c and d.

---

## Author Response (AR1)

**Response to reviewers**

We thank both reviewers for their useful comments on our paper. Please note that all line numbers refer to the version with *tracked changes*.

**Reviewer #1**

**Significant comments:**

However, I am very concerned whether their results are not unduly influenced - or even artefact - of the way they initialize their trajectory model.

If the initiation level is indeed 370K (and this is not a typo), then the model is essentially initialized with the driest possible state in the colder regions (over the cold Western Pacific, the tropopause is around 370K), and it is trivial that the model has a dry bias below that level - which is a problem particularly for the 100hPa level. (I should add that the display of data on pressure levels - necessary for MSL - makes it very difficult for the reader to understand whether the model is initialized above or below that pressure level.) Above 370K, this model may have trajectories that never experienced a true cold point, but the biggest problem is below that level: in this model, anything below 370K is populated only by descending pathways - and none have just directly come up from the troposphere (which can be expected to be at local saturation, and hence much moister). This most certainly biases the results strongly.

The serious concerns with the initialization preclude further discussion of the results since the quantitative numbers for convective moistening may be strongly affected. This paper tackles an important problem, but the authors need to demonstrate that their results are not unduly influenced by the initialization - which should be done in the troposphere and not at the tropopause. Major revisions are therefore inevitable, unfortunately.

This is a good point. We have re-run all the trajectory models with parcels initialized at 360 K. The 360-K potential temperature is above the average level of zero heating rates and below the cold point level, so parcels will ascend upon initialization and experience the cold point. For MERRA-2, the average heating rates below ~365 K in the NH subtropics are negative. To deal with this problem, we released those parcels above the level of zero heating rates for MERRA-2 simulations. See detailed information on our new initialization at 360 K (Section 2.4, Lines 218-255).

We have updated all our trajectory results (Figures 1-2 and Figures 5-9) and discussion accordingly throughout the paper.

While some quantitative details of the analyses have changed, our conclusion is unchanged: The trajectory models are still unable to simulate the seasonal cycle in the NH subtropics when water vapor is influenced only by temperature; Our simulations still show that convective ice evaporation is important for the NH subtropical seasonal cycle.

**Minor comments:**

Further - are the authors really initializing the trajectories in the extratropical stratosphere up to 60 degrees latitude with 200ppmv as stated in the text? This seems dramatically off - I am puzzled how this would produce the (reasonably looking) patterns shown in Figure 5?

The answer is, yes, we initialize parcels well into mid-latitudes. The reason these parcels do not impact the 100-hPa water values is that most of these mid- and high-latitude parcels are descending, so they never reach 100 hPa. Any that do ascend and advected into the TTL are dehydrated by cold TTL temperatures, so their water vapor values are set by TTL temperatures by the time they get to 100 hPa. We have edited the texts in the paper to make sure that there is no confusion (Lines 214-217 and Lines 254-255).

**Reviewer #2**

Since this is mostly diagnosing an output result from the GEOSCCM I dont see what the pupose of the model runs done with ERAi and MERRA-2 add to this. I think figures 1 and 2 could have been made from just using trajectories driven from GEOSCCM thus simplifying the analysis and figures. I assume that driving the trajectory model with any of these wind and temperature fields will produce similar results. Ie compare figure 5 to 1 and 2. Maybe I just missed something here.

We disagree with this comment. Without the plots showing ERAi- and MERRA2-based trajectory simulations, our paper would be a model-only analysis. We would not be able to connect the GEOSCCM results with observations. The fact that the results produced by ERAi and MERRA2 trajectory without convection look similar to those from the GEOSCCM trajectory without convection is a key link in the chain of logic of our paper. Thus, we will leave the figures in.

On page 11 line 325 it  $\rightarrow$  in.

We have updated the text. (Line 764)

**1 Impact of convectively lofted ice on the seasonal cycle of water vapor**

- 2 in the tropical tropopause layer
- 3 Xun Wang1, Andrew E. Dessler1, Mark R. Schoeberl2, Wandi Yu1, and Tao Wang3
- 4 1Department of Atmospheric Sciences, Texas A&M University, College Station, TX, USA
- 5 2Science and Technology Corporation, Columbia, MD, USA
- 6 3University of Maryland, College Park, MD, USA
- 7 Correspondence to: Andrew E. Dessler (adessler@tamu.edu)

Abstract. We use a forward Lagrangian trajectory model to diagnose mechanisms that produce water vapor seasonal cycle 8 observed by the Microwave Limb Sounder (MLS) and reproduced by the Goddard Earth Observing System Chemistry Climate 9 10 Model (GEOSCCM) in the tropical troppause layer (TTL). We confirm in both the MLS and GEOSCCM that the seasonal 11 cycle of water vapor entering the stratosphere is primarily determined by the seasonal cycle of TTL temperatures. However, we find that the seasonal cycle of temperature predicts a smaller seasonal cycle of TL water vapor between 10°N-40°N than 12 13 observed by MLSe or simulated by the GEOSCCM. Our analysis of the GEOSCCM shows that including evaporation of 14 convective ice in the trajectory model increases both the simulated maximum value of the 100-hPa 10°N-40°N water vapor 15 seasonal cycle as well as increasing the seasonal-cycle amplitude. We conclude that the moistening effect from convective ice 16 evaporation in the TTL plays a key role regulating and maintaining seasonal cycle of water vapor in the TTL. Most of the 17 convective moistening in the 10°N-40°N range comes from convective ice evaporation occurring at the same latitudes. A small contribution to the moistening comes from convective ice evaporation occurring between 10°S-10°N. Within the 10°N-40°N 18 band, the Asian monsoon region is the most important region for convective moistening by ice evaporation during boreal 19 20 summer and autumn

**21 **1 Introduction**

- 22 Stratospheric water vapor is important for the radiative budget of the atmosphere and the regulation of stratospheric ozone
- 23 (e.g., Solomon et al., 1986; Dvortsov and Solomon, 2001). One of the key features of the tropical lower stratospheric (LS)
- 24 water vapor is its seasonal cycle often referred to as the "tape recorder" (Mote et al., 1995, 1996). The amount of water vapor
- 25 entering the stratosphere and its seasonal cycle is primarily controlled by temperatures in the tropical tropopause layer (TTL)
- 26 (Brewer, 1949; Holton et al., 1995; Fueglistaler et al., 2009). The low TTL temperatures freeze-dry the air, reducing the
- 27 water vapor mixing ratios and imprinting the seasonal cycle on air ascending into the stratosphere through the TTL (e.g.,
- 28 Mote et al., 1996; Fueglistaler, 2005; Schoeberl et al., 2008; Fueglistaler et al., 2009).
- 29 Analyses of observations have suggested that deep convection reaching the TTL may also be important for regulating the

[revised manuscript text omitted]

| Deleted: tropical LS                                                                                                                                                    |
|-------------------------------------------------------------------------------------------------------------------------------------------------------------------------|
| Deleted: , however,                                                                                                                                                     |
| Deleted: The seasonal cycle is one of the key features of the tropical LS water vapor, so it is important that we fully understand the mechanisms that drive it. |
| Deleted: test whether moistening by                                                                                                                                     |
| Deleted: is playing a role                                                                                                                                              |
|                                                                                                                                                                         |

the TTL and LS. The CALIOP cloud Ice Water Content (IWC) is derived from a parameterized function of the CALIOP 532

nm cloud particle extinction profiles (Avery et al., 2012; Heymsfield et al., 2014). We use the IWC from all clouds minus the

IWC from thin cirrus clouds (clouds that are not opaque) above 146 hPa, which is a rough estimate of convective ice in the

- 194 TTL region, since the CALIOP does not separate convective from non-convective IWC measurements. These CALIOP IWC
- 195 data, obtained between May 2008 and December 2013, are then monthly averaged onto the same horizontal and vertical grids
- 196 as were used for the MLS data.

**197 2.3 GEOSCCM**

We also analyze simulations, from the Goddard Earth Observing System Chemistry Climate Model (GEOSCCM). The

- 199 GEOSCCM couples the GEOS-5 general circulation model (Rienecker et al., 2008; Molod et al., 2012) to a comprehensive
- 200 stratospheric chemistry module (Oman and Douglass, 2014; Pawson et al., 2008). The GEOSCCM uses a single-moment
- 201 cloud microphysics scheme (Bacmeister et al., 2006; Barahona et al., 2014). The run analyzed here starts in 1998 and ends in
- 202 2099 and driven by the Representative Concentration Pathway (RCP) 6.0 greenhouse gas scenario (Van Vuuren et al., 2011)
- 203 and the A1 scenario for ozone depleting substances (World Meteorological Organization, 2011). Sea surface temperatures
- and sea ice concentrations were prescribed from Community Earth System Model version 1 simulations (Gent et al., 2011).
- 205 The model has a horizontal resolution of 2°latitude by 2.5°longitude and 72 vertical levels up to the model top at 0.01 hPa
- 206 (Molod et al., 2012),

**207 2.4 Trajectory Model**

We also use the forward, domain filling, diabatic trajectory model described in Schoeberl and Dessler (2011) and updated in

- 209 subsequent publications. The trajectory model uses 6-hourly instantaneous horizontal winds and 6-hourly average diabatic
- 210 heating rates to advect parcels using the Bowman trajectory code (Bowman, 1993; Bowman and Carrie, 2002)
- 211 Meteorological fields used to drive the model in this paper come from the European Centre for Medium-Range Weather
- 212 Forecasts (ECMWF) ERA-interim (ERAi), and Modern-Era Retrospective analysis for Research and Applications-2
- 213 (MERRA-2) (Molod et al., 2015; Gelaro et al., 2017), and the GEOSCCM.
- In this study, the trajectory model initializes 1350 parcels daily in the upper troposphere on an equal area longitude-latitude
- 215 grid covering 0-360° longitude and  $\pm$  60° latitude, and with initial water vapor mixing ratio of 200 parts per million by volume
- 216 (ppmv). This value is well above saturation, so the parcels are dehydrated to saturation after the first time step of the
- 217 trajectory model run. Sensitivity tests show that our results are not impacted by the initialization values.
- The initialization level is at 360-K potential temperature, which is above the average level of zero heating (~355-360 K)
- (Fueglistaler et al., 2009) but below the tropical cold point. In the MERRA-2, the average heating rates below ~365 K in the
- 220 NH subtropics are negative during boreal summer (not shown), which results in parcels in that region immediately

**(Deleted: ice**

|  | Deleted: | detected | by | the | CALIOP |  |
|--|----------|----------|----|-----|--------|--|
|--|----------|----------|----|-----|--------|--|

- Deleted: In this study, we average
- Deleted: interpolate the CALIOP IWC into

[revised manuscript text omitted]

(Deleted: TTL and LS

To test if GEOSCCM convective ice field is realistic, we compare GEOSCCM convective ice with CALIOP ice data (ppmv)

(Figures 3 and 4). For the CALIOP, we show IWC from all clouds minus IWC from thin cirrus clouds (not opaque), above

146 hPa, which is a rough estimate of convective ice in the TTL region, although it is almost certainly an underestimate of true convective ice amount. There's general agreement between the spatial pattern of GEOSCCM and CALIOP convective ice.

However, the GEOSCCM generally produces more convective ice and higher convective top altitudes than the CALIOP, To address these problems in the GEOSCCM, we also show the GEOSCCM convective ice field reduced by 80% (0.2ice), which brings tropical GEOSCCM convective ice into better agreement with the CALIOP values at 121 hPa and above (Figs. 3e-f and

4e-f). We show below two sensitivity tests that show our results are not sensitive to the overestimation of convective IWC and convective top altitude by the GEOSCCM

The water vapor seasonal cycles from the GEOSCCM and various GEOSCCM trajectory model runs (Table 1) are shown in Fig. 5. These have been re-averaged in the vertical using the MLS averaging kernels (Livesey et al., 2017) to facilitate comparison with MLS. We focus on the 100-hPa level, where the hemispheric asymmetry is strongest. We note that the 100hPa level is in the TTL and is not strictly above the tropopause, especially in the summer NH monsoon region. However, processes on this level play a key role in determining stratospheric water vapor (Fueglistaler et al., 2009).

The GEOSCCM reproduces the hemispheric asymmetry seen in the MLS observations (compare Fig. 5a with Fig. 1c), and shows that during JJA the 100-hPa water vapor maxima are located over the Asian monsoon and North American monsoon regions (compare Fig. 5b with Fig. 2c). The standard trajectory model driven by GEOSCCM meteorology, which regulates water entirely through TTL temperatures, does not reproduce the hemispheric asymmetry (Fig. 5c). That model also underestimates the JJA water vapor values in the Asian monsoon region and North American monsoon region (Fig. 5d). These results are similar to the comparison between MLS and the standard trajectory models driven by ERAi and MERRA-2 (Figs. 1f, 1i, 2f, and 2i)

Fig. 6 shows the 100-hPa water vapor seasonal cycles in the NH subtropics (10°N-40°N), deep tropics (10°S-10°N), and 426 southern hemispheric subtropics (10°S-40°S). To aid in comparison, we have subtracted the annual mean from each data set. 427 The standard model generally agrees well with the GEOSCCM and MLS in the 10°S-10°N and 10°S-40°S region (Figs. 6b-428 d). This suggests that the water vapor seasonal cycle in those regions is mainly controlled by the TTL temperatures and large-429 scale transport and implying that other factors, including convective ice evaporation, are less important. In the 10°N-40°N 430 region, however, the standard model does a poor job, underestimating the MLS and GEOSCCM seasonal amplitude by 1.15 431 ppmv (55%) and 1.23 ppmv (57%) (Figs. 6a and 6d).

If we add convective ice evaporation to the trajectory model, then the models show a clear hemispheric asymmetry in the 100- hPa water vapor seasonal cycle and more pronounced seasonal maxima over the monsoon regions (Figs. 5e-h). Fig. 6 shows that the ice model and the 0.2 ice model (the trajectory model where we add 0.2 ice as shown in Figs. 3e-f) produce boreal

| 1 | Deleted: I | n Figures 3 and 4 |  |
|---|------------|-------------------|--|
|   |            |                   |  |

**Deleted: IWC**

**Deleted: convective IWC**

**Deleted: convective IWC.**

(Deleted: Figs. 5a-b).

**Deleted: Figs. 5c-d**

[revised manuscript text omitted]
. Deleted: absolute values of Deleted: do Deleted: subtract Deleted: but during Deleted: 46 Deleted: (71%) Deleted: 1.09 Deleted: (86%) Deleted: Deleted: 50 Deleted: (24.7%) Deleted: 12 Deleted: (9.9%) of the net convective moistening in the 10°N-40°N water vapor seasonal maximum value and seasonal amplitude. Deleted: calculate

 $\begin{array}{l} 635 \quad 10^{\circ}\text{N}-40^{\circ}\text{N} \text{ and } 10^{\circ}\text{S}-10^{\circ}\text{N}. \text{ Fig. 9 shows the contribution from each box region to the net convective moistening in the 100-$  $636 \quad \underline{\text{hPa}}_{1}10^{\circ}\text{N}-40^{\circ}\text{N} \text{ water vapor seasonal maximum value in September and the seasonal amplitude.} \end{array}$

We find that contribution from the box regions over Southeast Asia (10°N-40°N, 60°E-120°E), subtropical Western Pacific 637 638 (10°N-40°N, 120°E-180°E), and North America (10°N-40°N, 120°W-60°W) dominate. The Southeast Asia region is most 639 important, contributing to 20% (0.54 ppmv) and 20% (0.2 ppmv) of the net convective moistening in the 10°N-40°N water 640 vapor seasonal maximum value and seasonal amplitude, respectively. This conclusion is consistent with Ueyama et al. (2018), 641 who showed that parcels in the 10°S-50°N domain at 100 hPa are mainly hydrated by convection over Southeast Asia. 642 Specifically, they showed that convection over the Asian monsoon region (0-40°N,40°E-140°E) contributes approximately 50% of the total convective moistening (10°S-50°N) at 100 hPa during August 2007. We computed the contribution from the 643 same domain and got a contribution of 36%. The reason we produce a smaller contribution from this domain is that the 644 645 GEOSCCM produces more convective ice over the tropical west Indian Ocean (Fig. 4), which results in larger convective 646 moistening contributed by that region.

The subtropical Western Pacific also contributes to the net convective moistening in the 100-hPa 10°N-40°N water vapor 648 seasonal cycle. This is due to the abundant convective ice over the subtropical west Pacific (Fig. 4b), which is likely related to the east-west oscillation of the Asian monsoon anticyclone (Pan et al., 2016; Luo et al., 2018). The North America region is less important in the ice model, contributing to 12% (0,2 ppmv) and 13% (0,21 ppmv) of the net convective moistening in the

10°N-40°N water vapor seasonal maximum value and seasonal amplitude. The GEOSCCM underestimates the observed convective ice over the North American monsoon above 120 hPa (not shown), which may cause the contribution from the

North American region to be underpredicted.

**654 4.Summary**

In this study, we investigated mechanisms that drive the seasonal cycle of water vapor in the TTL. We use a Lagrangian trajectory model (Schoeberl and Dessler, 2011) to analyze the seasonal cycle in observations of water vapor made by the algeboly moder (behoever) and Dessier, 2011) to analyze the seasonal open in observations of water rapor made by the

Microwave Limb Sounder (MLS) (Lambert et al., 2007; Livesey et al., 2017) as well as simulated fields from the Goddard

- Earth Observing System Chemistry Climate Model (GEOSCCM) (Rienecker et al., 2008; Molod et al., 2012; Pawson et al.,
- 659 2008; Oman and Douglass, 2014).

Water vapor's seasonal cycle in the TTL and tropical lower stratosphere (LS), sometimes referred to as the "tape recorder,"

has highest values of water vapor entering the stratosphere during NH summer. We confirm in both the MLS observations and in the GEOSCCM that this is mainly due to the seasonal cycle of TTL temperatures. However, closer examination of the data reveals some deficiencies in this simple picture. Both the MLS and GEOSCCM show that the water vapor seasonal cycle in the TTL has a hemispheric asymmetry, with maximum seasonal cycle between 10°N-40°N, despite the fact that the TTL

[revised manuscript text omitted]

| Deleted: maximum value of the                                       |  |
|---------------------------------------------------------------------|--|
| Deleted: water vapor                                                |  |
| Deleted: cycle                                                      |  |
| Deleted: 9                                                          |  |
| Deleted: 47%)                                                       |  |
| Deleted: 10°N-40°N                                                  |  |
| Deleted: 26                                                         |  |
| Deleted: 123%) compared to standard model. ¶
A major part of the |  |
| Deleted: 10°N-40°N water vapor seasonal cycle                       |  |

| -( | Deleted: IWC                                       |
|----|----------------------------------------------------|
| -( | Deleted: 1.46 ppmv (71%)                           |
| (  | Deleted: 1.09 ppmv (86%)                           |
| (  | Deleted: is                                        |
| (  | Deleted: region for convective ice evaporation and |

tropics between 10°S-10°N, has a smaller influence in 100-hPa water vapor between 10°N-40°N. However, since the 756

GEOSCCM underestimates the observed convective ice over the North American monsoon above 120 hPa (not shown), it is likely that this causes an underestimation of the moistening effect of convective ice over the North American region. Previous studies showed that the ratio of isotopic water vapor (HDO), an indicator of sublimation of convective ice and in-mixing (e.g.,

Dessler et al., 2007; Hanisco et al., 2007; Randel et al., 2012), enhances over the American monsoon region during boreal summer, suggesting more convective ice evaporation there (Randel et al., 2012). This paper does not discuss the HDO issue, 761

and more work needs to be done in the future.

To summarize, we find that TTL temperature variations alone cannot explain the seasonal cycle of water vapor at 100 hPa in

MLS observations over the NH subtropics, 10°N-40°N (although temperature does explain the seasonal cycle in the tropics, 764

10°S-10°N and southern subtropics, 10°S-40°S). To try to understand the other mechanisms at work, we analyze a chemistry- climate model, the GEOSCCM, which reproduces the MLS observations and has been shown to accurately simulate the TTL.

"We find that, in the GEOSCCM, evaporation of convective ice in the TTL is responsible for the Jarger seasonal cycle in the

J00-hPa NH subtropics. We therefore conclude that evaporation of convective ice in the TTL, mainly in boreal summer, is the most likely explanation for the observed larger seasonal cycle in the NH subtropics. We concur that the seasonal cycle of the

TTL temperatures is the major driver of the seasonal cycle of water vapor entering the stratosphere, but we find that the contribution from evaporation of convective ice fills in more details of this simple picture. Our findings emphasize the need to better understand and quantify the magnitude and spatial pattern of convective ice evaporation in the TTL.

Data availability. The water vapor observed by MLS is available from https://mls.jpl.nasa.gov/. The ice water content observed by CALIOP is available from https://eosweb.larc.nasa.gov/. The MERRA-2 meteorological fields are available 774

from https://disc.gsfc.nasa.gov/. The ERAi meteorological fields are available from https://www.ecmwf.int/en/forecasts/datasets/reanalysis-datasets/era-interim.

Competing interests. The authors declare that they have no conflict of interest.

Author contribution. Xun Wang performed analysis, and wrote the original draft. Andrew E. Dessler provided the conceptualization, guidance, and editing. Mark R. Schoeberl and Tao Wang contributed to the trajectory model code, 779

methodology, discussion, and editing. Wandi Yu contributed to methodology and discussion.

Acknowledgments. This work was supported by NASA grants NNX16AM15G and 80NSSC18K0134, both to Texas A&M 781 782 University. We would like to thank Dr. Luke Oman for providing the GEOSCCM meteorological fields used in this study.

[revised manuscript text omitted]